# Microtubule plus-end dynamics link wound repair to the innate immune response

**Clara Taffoni[1†], Shizue Omi[1], Caroline Huber[1], Sébastien Mailfert[1], Mathieu Fallet[1], Jean-François Rupprecht[2], Jonathan J Ewbank[1], Nathalie Pujol[1]\***

[1]CIML, Centre d'Immunologie de Marseille-Luminy, Turing Centre for Living Systems, Aix Marseille Univ, INSERM, CNRS, Marseille, France; [2]CPT, Turing Centre for Living Systems, Aix Marseille Univ, CNRS, Marseille, France

**Abstract** The skin protects animals from infection and physical damage. In *Caenorhabditis elegans*, wounding the epidermis triggers an immune reaction and a repair response, but it is not clear how these are coordinated. Previous work implicated the microtubule cytoskeleton in the maintenance of epidermal integrity (Chuang et al., 2016). Here, by establishing a simple wounding system, we show that wounding provokes a reorganisation of plasma membrane subdomains. This is followed by recruitment of the microtubule plus end-binding protein EB1/EBP-2 around the wound and actin ring formation, dependent on ARP2/3 branched actin polymerisation. We show that microtubule dynamics are required for the recruitment and closure of the actin ring, and for the trafficking of the key signalling protein SLC6/SNF-12 toward the injury site. Without SNF-12 recruitment, there is an abrogation of the immune response. Our results suggest that microtubule dynamics coordinate the cytoskeletal changes required for wound repair and the concomitant activation of innate immunity.

**\*For correspondence:**
pujol@ciml.univ-mrs.fr

**Present address:** [†]Institute of Human Genetics, University of Montpellier, CNRS, Montpellier, France

**Competing interests:** The authors declare that no competing interests exist.

## Introduction

Inducible immune responses are ubiquitous features of animal defences against infection. In one well-studied example, when the fungus *Drechmeria coniospora* infects *Caenorhabditis elegans*, by piercing the cuticle and growing through the underlying epidermis, it triggers an innate immune response, characterized by the induction of expression of a battery of antimicrobial peptide (AMP) genes (*Pujol et al., 2008b*; *Taffoni and Pujol, 2015*). These include the genes of the *nlp-29* cluster (*Pujol et al., 2012*), which are principally controlled in a cell autonomous manner (*Lee et al., 2018*). Sterile wounding also upregulates the expression of the *nlp-29* cluster (*Pujol et al., 2008a*). In both cases, these changes in gene expression result from the activation of the GPCR DCAR-1 that acts upstream of a conserved p38 MAPK signalling cassette (*Zugasti et al., 2014*). The p38 MAPK PMK-1 itself acts upstream of the STAT transcription factor-like protein STA-2. Although the precise details are as yet unclear, activation of STA-2 is believed to require the SLC6 family protein SNF-12 (*Dierking et al., 2011*).

Triggering the immune response upon wounding has a prophylactic role and is a safeguard against the potential entry of pathogenic microbes. To ensure longer term protection, a wounded tissue must also be repaired to maintain organismal integrity. Thus, in addition to AMP gene induction, skin wounding in *C. elegans* also triggers a Ca$^{2+}$-dependent signalling cascade that promotes wound repair through actin ring closure (*Xu and Chisholm, 2011*). The nematode skin is mainly a syncytium. Except at the head and tail, a single multinucleate epidermal cell, hyp7, covers the whole animal (*Altun and Hall, 2014*). Its repair processes exhibit some of the characteristics previously described for single cell wound repair in other organisms (*Sonnemann and Bement, 2011*). On the

other hand, actin ring closure in *C. elegans* does not rely on myosin-II contractility and the 'purse string' mechanism described in other organisms (*Begnaud et al., 2016*), but instead requires Arp2/3-dependent actin polymerization (*Xu and Chisholm, 2011*).

A small number of reports have indicated that microtubules (MTs) cooperate with the actomyosin cytoskeleton during wound closure. In a single-cell wound-healing model using *Xenopus laevis* oocytes, MTs were shown to be required to restrict the assembly zone of actin and myosin around the wound edge (*Bement et al., 1999*; *Mandato and Bement, 2003*). A study in *D. melanogaster* embryos also revealed that perturbation of MT dynamics, in a + end-binding protein 1 (EB1) mutant, resulted in a delay of actomyosin assembly at the wound edge in multi-cellular wounds (*Abreu-Blanco et al., 2012*). The direct role of MTs in the *C. elegans* wound response has not been previously addressed.

The tissue repair processes in *C. elegans* have been described to act in parallel to the innate immune response that accompanies wounding (*Xu and Chisholm, 2011*). Several pieces of evidence suggest, however, a coordination of the two responses. Earlier work had shown that loss of DAPK-1, homolog of death-associated protein kinase, causes both undue tissue repair and an inappropriate activation of an immune response (*Tong et al., 2009*). A more recent study showed that loss of *dapk-1* function causes excessive MT stabilization (*Chuang et al., 2016*). These results suggest that MT stability could directly influence immune signalling.

Here, we develop the use of a 405 nm laser to wound the *C. elegans* epidermis and characterise the subsequent subcellular events in vivo. We show for the first time a rapid membrane reorganisation of phosphatidylinositol 4,5-bisphosphate ($PIP_2$) domains, as well as a recruitment of EB1 and reorganisation of MTs around the wound. Rings of EB1 and actin close at the same pace, with EB1 at the leading edge. We demonstrate that specific inactivation in the adult epidermis of several alpha and beta tubulin isotypes leads to a decrease in MT dynamics, revealed by an absence of EB1 comets. This inactivation also blocks EB1 accumulation and decreases the recruitment of actin at the wound site. Further, it also abrogates the directed trafficking to the wound site of the key signalling protein SNF-12 and causes a block of the subsequent immune response. These results suggest that the recruitment of MT + end and EB1 to wounds coordinates the activation of the innate immune response and wound repair.

## Results

### The *C. elegans* epidermis can be wounded with a 405 nm laser

The adult epidermis in *C. elegans* is a single cell layer, covered on its apical surface by an impermeable cuticle that serves as an exoskeleton. The epidermal syncytium hyp7 can be divided into two distinct regions, one lateral, where the nuclei and most of the cytoplasm are, and the dorso-ventral part above the muscles where the apical and basolateral plasma membranes of hyp7 are juxtaposed and crossed by hemidesmosomes that anchor the muscles to the cuticle exoskeleton (*Figure 1A*; *Altun and Hall, 2014*). The large size of hyp7 makes it an attractive model for the study of cellular wound repair mechanisms (*Pujol et al., 2008a*; *Xu and Chisholm, 2011*). Previous studies on *Xenopus* oocytes have demonstrated that the plasma membrane can be perforated in an extremely targeted manner using standard 405 nm microscopy imaging lasers (*Burkel et al., 2012*; *Mandato and Bement, 2001*). We found that we could efficiently wound the hyp7 lateral syncytial epidermis of *C. elegans* (*Figure 1A*) with a short pulse of 405 nm laser light (maximum power for 1–3 s) in Fluorescence Recovery After Photobleaching (FRAP) mode. Coupled with spinning disk or confocal microscopy, it provides a reproducible way to monitor in vivo the immediate consequences of a carefully controlled injury and the subsequent steps in the wound healing process. Using this system, we first confirmed all the previously described wound hallmarks (*Pujol et al., 2008a*; *Xu and Chisholm, 2011*), such as an immediate autofluorescent scar, a $Ca^{2+}$ burst, the formation of an actin ring and later the induction of AMP reporter gene expression (*Figure 1—figure supplement 1*).

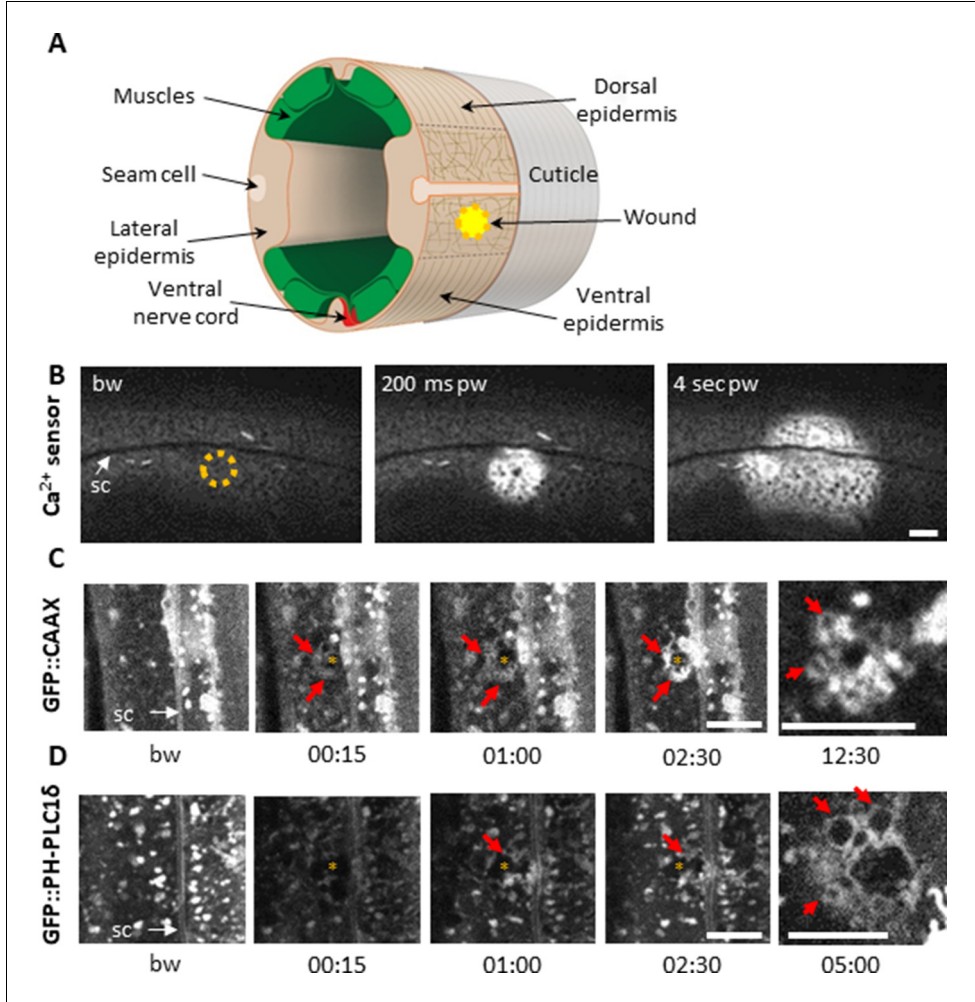

**Figure 1.** Using a 405 nm laser to wound the worm epidermis reveals rapid membrane reorganisation. (**A**) Schematic view of a section through an adult *C. elegans* worm near the mid-body. The internal organs are omitted for the sake of clarity. The main syncytial epidermis, hyp7 (buff), with its adjacent cells (seam cells in pink, nerve cord in red and muscles in green) can be divided into two main regions, the lateral epidermis (delimited by the dashed black line) and the thin dorso-ventral epidermis where muscles are anchored to the cuticle. Microtubules (thin taupe lines) are disordered on the apical surface of the lateral epidermis. The typical position of a wound is indicated by the yellow circle. Figure adapted from one kindly provided by Christopher Crocker, WormAtlas (**Altun and Hall, 2014**). (**B**) Wounding the lateral epidermis with a 405 nm laser causes a rapid increase in intracellular $Ca^{2+}$, measured using GCamp3. Representative spinning disk images from a worm carrying a *col-19p:: GCamp3* reporter transgene. The dashed circle is centred on the wound site; bw, before wound; pw, post-wound; sc, seam cells. Scale bar 10 µm. (**C–D**) Upon laser wounding, membrane and lipids (red arrows) are rapidly recruited to the wound site (asterisk). Representative spinning disk images of worms carrying *dpy-7p::GFP::CAAX* (C) and *wrt-2p::GFP::PH-PLC1δ* transgenes (**D**). bw, before wound; time post-wound [min:s]; sc, seam cells. Scale bar 5 µm. The last panel on the right in both C and D is a focal plane two microns deeper than the previous images, showing recruitment of cytoplasmic vesicles to the wound.

The online version of this article includes the following video and figure supplement(s) for figure 1:

**Figure supplement 1.** Reproduction of known wound hallmarks following injury using a 405 nm laser.

**Figure 1—video 1.** Plasma membrane rapidly reorganizes at the wound site.

https://elifesciences.org/articles/45047#fig1video1

**Figure 1—video 2.** Rapid PIP$_2$ domain reorganization at the wound site.

https://elifesciences.org/articles/45047#fig1video2

## Membrane is rapidly reorganised at the wound site

In *Xenopus* oocytes, a wound is rapidly patched through mobilisation and recruitment of membrane from a local pool of vesicles (*Davenport et al., 2016*). We could similarly observe rapid membrane recruitment at the wound site in the epidermal syncytium using a strain expressing a prenylated form of GFP (*dpy-7p::GFP::CAAX*). Within 15 s after wounding, large heterogeneous membrane domains that are normally stable regrouped and extended towards the wound (*Figure 1C*, *Figure 1—video 1*). PIP$_2$ has been found segregated into distinct membrane pools in the plasma membrane in other species, consistent with its wide range of cellular functions including regulating the adhesion between the actin-based cortical cytoskeleton and the plasma membrane (*Raucher et al., 2000*). We found that the plasma membrane of the hyp7 syncytium was also heterogeneously labelled using a strain in which PIP$_2$ was specifically labelled (*wrt-2p::GFP::PH-PLC1δ Wildwater et al., 2011*); these membrane domains were relatively static. Upon wounding, we observed a rapid but transient disappearance of the signal in a larger area than the wound. There was then a reorganization of these membrane domains around the wound after 15 to 30 s, followed by appearance of cytoplasmic vesicles below the plasma membrane (*Figure 1D*, *Figure 1—video 2*). These processes are likely to provide an immediate barrier preventing the potential leakage of cellular components, followed by a restoration of membrane integrity as previously proposed in other systems (*Davenport et al., 2016*; *Nakamura et al., 2018*; *Vaughan et al., 2014*).

## Actin is recruited at the wound site forming concentric rings

Wounding the *C. elegans* epidermis causes cytoskeleton rearrangements, with an actin ring forming at the wound site (*Xu and Chisholm, 2011*). We wanted to monitor the dynamics of actin recruitment and so constructed a strain in which the filamentous actin binding peptide Lifeact (*Riedl et al., 2008*), labelled with mKate2, was specifically expressed in the adult epidermis (*col-62p::Lifeact::mKate2*). Under resting condition in the young adult, in the lateral epidermis, we observed a disorganised pattern of foci linked with fine filaments (*Figure 2A*, bw (before wound)). This is in contrast to the highly organised circumferential bundles of actin found above the muscles in the same hyp7 cell, as previously described (*Costa et al., 1997*; *Lažetić et al., 2018*; *Figure 2—figure supplement 1A*). Upon wounding, while the pre-existing actin filaments appeared stable, actin was recruited as a wave to the wound site within 2–4 min and formed a well-defined actin ring after 5 min (n > 100). This ring subsequently constricted as the wound closed (*Figure 2A* and *Figure 2—video 1*), in line with previous observations (*Xu and Chisholm, 2011*).

In the adult epidermis, wounds do not close by a purse-string mechanism but by local actin polymerization involving the Arp2/3 complex (*Xu and Chisholm, 2011*). The Arp2/3 complex produces branched actin networks, and it contains in *C. elegans* seven subunits, encoded by *arx-1* through *arx-7* (*Sawa et al., 2003*). To confirm its role in the formation of the actin ring, we examined ARX-2/ARP2 during wounding, using an ARX-2::GFP knock-in strain (*Wu et al., 2017*). We could observe a recruitment of ARX-2/ARP2 to the wound with a pattern similar to actin (*Figure 2B*, *Figure 2—video 2*). These results reinforce the hypothesis that the actin ring closes due to polymerisation of branched actin network.

## EB1 accumulates at the wound site with actin

Microtubules (MTs) are another essential component of the cytoskeleton, but their role in wound healing has not previously been addressed directly in *C. elegans*. To look at MT dynamics, we imaged worms expressing a tagged form of EB1/EBP-2, a protein that binds to the + end of growing MTs (*Akhmanova and Hoogenraad, 2005*; *Srayko et al., 2005*), specifically in the adult epidermis (*col-19p::EBP-2::GFP*; EB1::GFP for brevity) (*Chuang et al., 2016*). Under resting conditions EB1::GFP exhibited rapid, comet-like movement throughout hyp7, with a speed (0.26 ± 0.10 μm/s, *Figure 2—source data 1*) similar to the speed previously determined (0.3 μm/s) (*Chuang et al., 2016*). Upon wounding, EB1::GFP comets accumulated within 2 min around the wound site (n > 50, *Figure 2C* and *Figure 2—video 3*). This was concomitant with a small reduction of their average speed in the vicinity of the wound, to 0.16 ± 0.06 μm/s (*Figure 2—source data 1*). When actin and EB1 were visualised simultaneously, EB1 was found associated with the actin ring as it closes, with a preferential localisation at the leading edge (*Figure 2D*, *Figure 2—videos 4* and *5*). Using a dynactin reporter previously shown to be associated with MT + tips in *C. elegans*, DNC-2::GFP

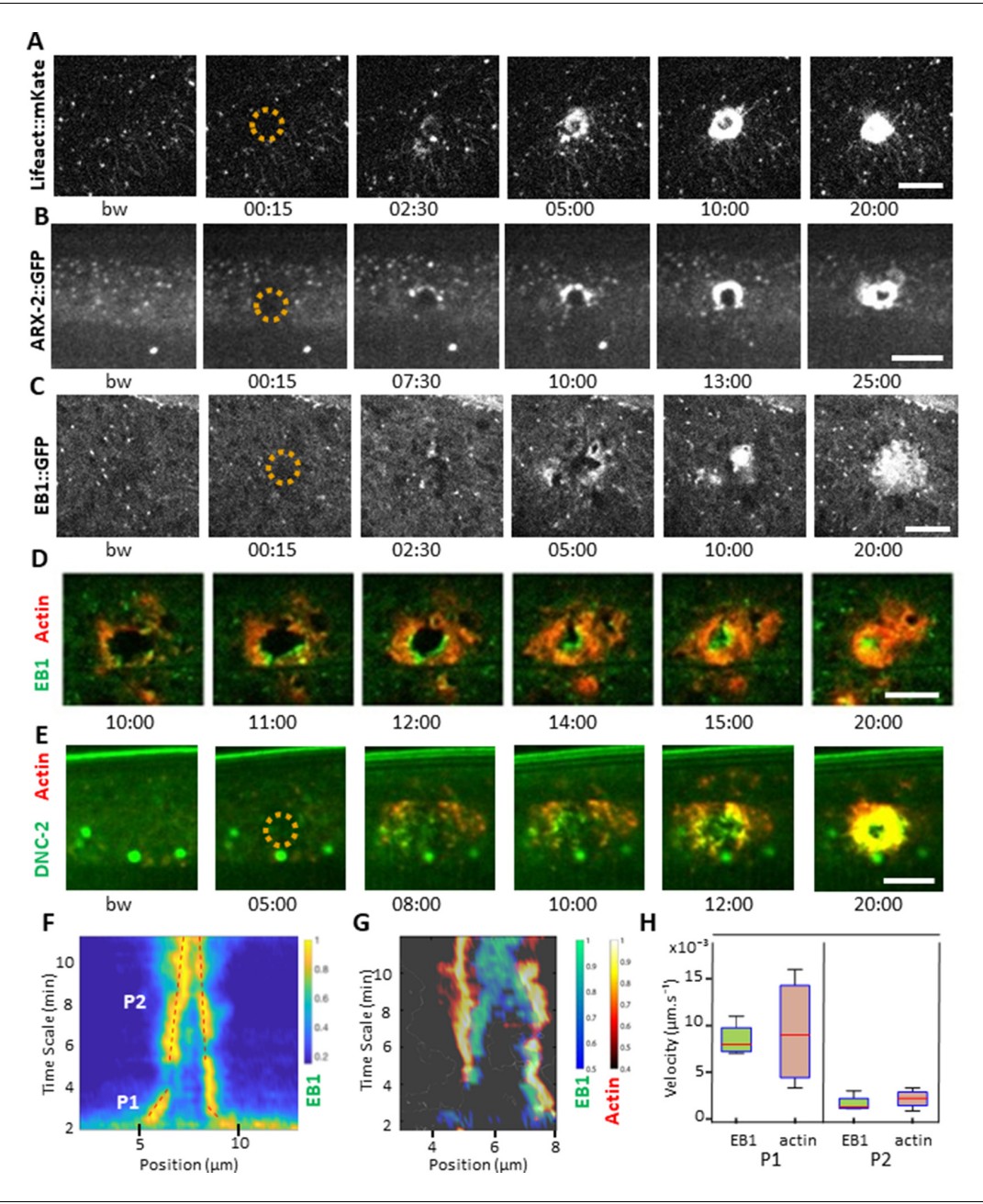

**Figure 2.** EB1 associates with actin at the wound site in the lateral epidermis. (**A**) In unwounded worms, in the lateral epidermis actin is sparsely structured. Laser wounding causes actin recruitment (as early as 2.5 min) and actin ring formation (clearly visible by 5 min). These rings close with time. Representative spinning disk images of a strain carrying a *col-62p::Lifeact::mKate* reporter. Laser wounding also causes ARP2/ARX-2 and EB1 recruitment at the wound site as visualized in strains ARX-2::GFP KI (**B**) and *col-19p::EBP-2::GFP* (**C**), respectively. Simultaneous visualization of MTs (green) and actin (red) using strains carrying EB1::GFP (**D**) or DNC-2::GFP (**E**) together with *col-62p::Lifeact::mKate2*. The dashed circle is centred on the wound site; bw, before wound; time post-wound [min:s]; scale bar 10 μm. (**F**) Kymograph of EB1 signal along the anterio-posterior (AP) direction normalised according to maximal intensity, see Materials and methods, reveals two phases of fast (P1) and slow (P2) contraction. (**G**) Kymograph of EB1 (green) and actin (red) during wound closure, EB1 ring contraction precedes that of actin. (**H**) Velocities of contraction in the fast (P1) and slow phases for EB1 and actin (n = 4).

The online version of this article includes the following video, source data, and figure supplement(s) for figure 2:

**Source data 1.** Reporter protein dynamics in the epidermis.

**Figure supplement 1.** Cytoskeleton organisation and dynamic in the lateral epidermis.

*Figure 2 continued on next page*

*Figure 2 continued*

**Figure supplement 2.** EB1 precedes actin during wound closure.

**Figure 2—video 1.** Actin reorganization at the wound site.

https://elifesciences.org/articles/45047#fig2video1

**Figure 2—video 2.** ARP2/3 complex is recruited at the wound site.

https://elifesciences.org/articles/45047#fig2video2

**Figure 2—video 3.** EB1/EBP-2, a plus end MT binding protein, is recruited at the wound site.

https://elifesciences.org/articles/45047#fig2video3

**Figure 2—video 4.** EB1 and actin are recruited at the wound site.

https://elifesciences.org/articles/45047#fig2video4

**Figure 2—video 5.** EB1 and actin are recruited at the wound site.

https://elifesciences.org/articles/45047#fig2video5

**Figure 2—video 6.** Kymograph analyses of EB1 and actin at the wound site.

https://elifesciences.org/articles/45047#fig2video6

---

(*Barbosa et al., 2017*), we observed that there was a similar recruitment of DNC-2 at the inner edge of the actin ring (*Figure 2E*, *Figure 4—video 1*).

To address further the nature of the association of EB1 with actin, we quantified the respective speeds of their accumulation during ring closure. Kymograph analyses of the EB1 signal revealed two phases, the first one almost 10 times more rapid that the subsequent one (*Figure 2F*). Simultaneous analyses of both EB1 and actin signals revealed the existence of two regimes for both (*Figure 2G*). Notably, the speed of EB1 ring closure (0.008 µm/s) was almost 2 orders of magnitude slower than the speed of the EB1 comets (0.26 µm/s), and was similar to that of actin ring closure (*Figure 2H*, n = 4). Moreover, the specific enrichment of EB1 at the leading edge and EB1 ring closure was ahead of that of actin (*Figure 2D and G*, *Figure 2—video 6*, *Figure 2—figure supplement 2*). These results show that EB1 associated with actin at the time of the formation of the ring up to its closure.

## MT dynamics at the wound

As EB1 comets reflect MT + tip dynamics, we then directly looked at the microtubule network after wounding. We used strains where one of the 16 *C. elegans* tubulin proteins, TBB-2, was labelled with GFP (*Si [col-19p::GFP::TBB-2]* and *KI [tbb-2p::TBB-2::GFP]*; both here termed TBB-2::GFP). We observed that MTs were arranged predominantly in longitudinal but haphazard bundles in the lateral epidermis, in marked contrast to their ordered, parallel and circumferential organisation underneath the muscle quadrants in the dorsal and ventral epidermis as well as during molting (*Figure 2—figure supplement 1B–D*), as previously described (*Costa et al., 1997*; *Wang et al., 2015*). Consistent with previous reports, EB1 comets were observed to run along MT bundles, with a preferred directionality in the A/P axis as observed for MT bundles in the main epidermis (*Figure 2—figure supplement 1E–G*). Tracking of EB1 comets confirmed this observation (*Figure 3A and B*). After wounding, no change in the direction of EB1 comets was observed (*Figure 3B*). Detailed observations with confocal microscopy revealed MT tips growing and retracting within second at the site of injury (*Figure 3C* and *Figure 3—video 1*). The dynamics of the MT tips paralleled the arrival of EB1 at the wound. At later time points, a global reorganisation of the MT network was observed as a knitting web with orthoradial MTs around the wound (*Figure 3D*, *Figure 3—videos 2* and *3*). The orientation of the MTs was analysed using a MAP1/MAPH-1.1::GFP reporter strain by quantifying the direction of the largest eigenvector of a local intensity structure tensor. Maps of their local orientation were defined within three concentric regions (R1-R3). The angle θ between the local direction of microtubule bundles and the direction to the centre of the wound region was then estimated (*Figure 3E*). After wounding, microtubules in the immediate wound periphery (R1) were perpendicular to the wound centre direction (θ ≈ 80). Such preferential orientation didnot exist in controls, for example in the distant sector R3 to the wound centre nor in any sectors R1-3 before laser ablation (*Figure 3F*). Thus, in concordance with the apparent random orientation of EB1 comets, these results show that MT were not arranged radially towards the wound, but displayed an increased orthoradial orientation proximal to the wound after 10 min. Together, these results lead to a working model in

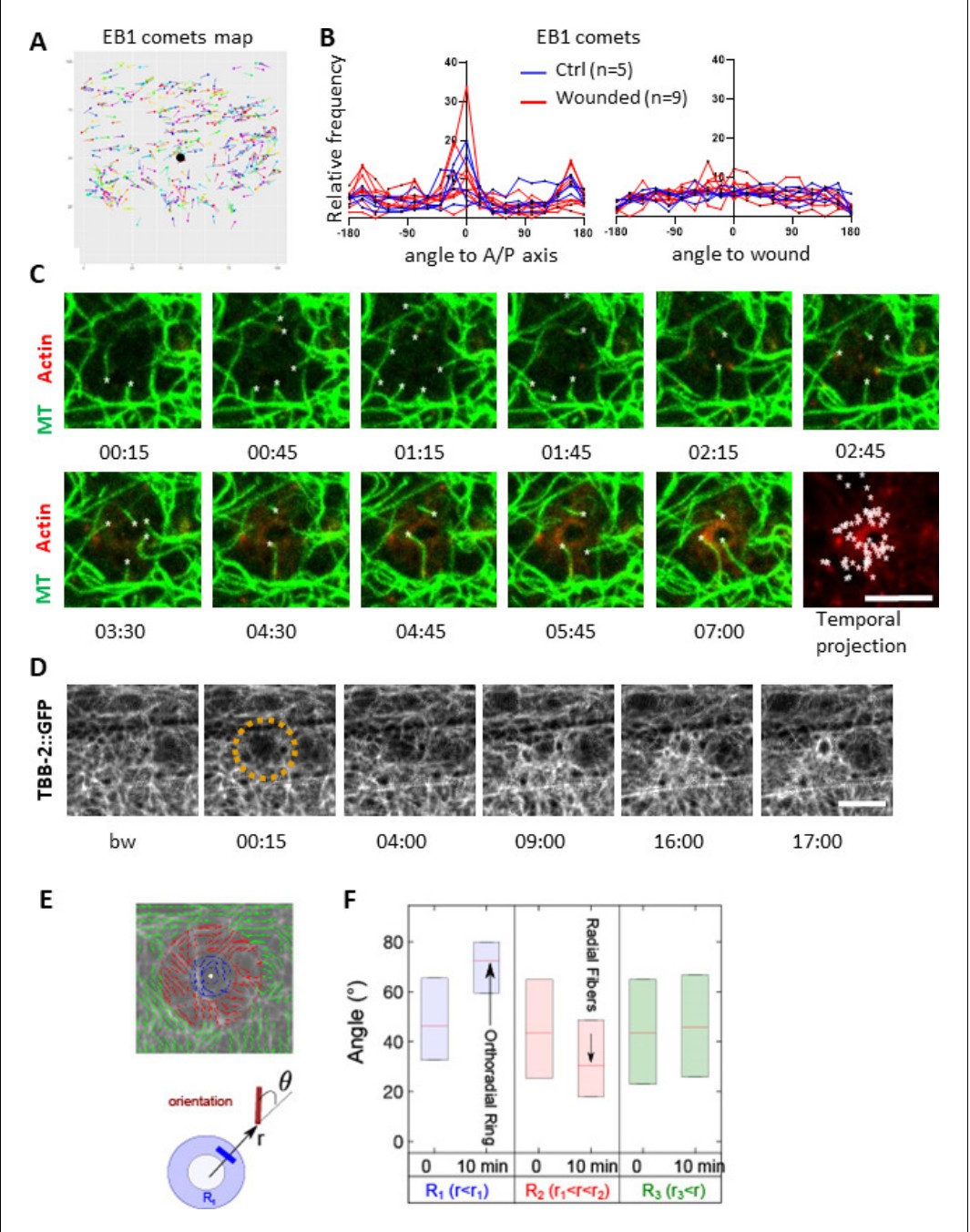

**Figure 3.** Microtubules are dynamic and orthoradially organised at the wound site. (A) EB1 comets are tracked and mapped in a region of 150 × 150 pixels centred on the wound (black dot) for 5 min. (B) The angles of the EB1 comet vectors are calculated either to the A/P axis of the worm (left) or to the wound or the centre of a control region (right); their relative frequency are compared between control (n = 5) and wounded regions (n = 9). (C) Representative confocal images of a strain carrying a *col-62p::Lifeact::mKate2* and *TBB-2::GFP* reporter; time post-wound [min:sec]; scale bar 5 μm. An asterisk marks the position of the tips of MTs extending into the site of the wound. The final panel is a temporal projection (*Figure 3—video 1*). (D) Representative spinning disk images of a strain carrying a *TBB-2::GFP* reporter. The dashed circle is centred on the wound site; time post-wound [min:sec]; scale bar 5 μm. (E) Map of the orientation of MTs 10 min post wounding, coloured according to their distance to the wound centre, defining sectors R1 (blue), R2 (red), R3 (green) in a MAPH-1::GFP reporter strain. Definition of the angle θ between the local direction of MT and the direction to the wound centre. (F) Statistics of the orientation θ for each sector just before (t = 0 min) and after (10 min) wounding. The increase in the mean value of

*Figure 3 continued on next page*

*Figure 3 continued*

θ in R1 corresponds to orthoradial MTs; the decrease in angle in R2 corresponds to the appearance of radially oriented structures; no global structures were observed at longer distances R3 (n = 3).

The online version of this article includes the following video(s) for figure 3:

**Figure 3—video 1.** Microtubules regrow around the wound site in front of actin.
https://elifesciences.org/articles/45047#fig3video1

**Figure 3—video 2.** Microtubules knit a web around the wound site.
https://elifesciences.org/articles/45047#fig3video2

**Figure 3—video 3.** Imaging KI MAPH-1::GFP strain; * indicates the wound site; [min:s].
https://elifesciences.org/articles/45047#fig3video3

which MTs rapidly seed the accumulation of EB1, so facilitating the recruitment of actin, as well as knitting a webbed network around the site of injury.

## *tba-2* or *tbb-2* inactivation results in abrogation of EB1 dynamics in the epidermis

To test whether MT growth at the wound site indeed influences actin ring formation, we sought a way to block MT dynamics prior to wounding the epidermis. Due presumably to their impermeable cuticle, treatment of adult worms with commonly used drugs, such as colchicine, was either ineffective or had irreproducible effects in our hands. Although inactivating various genes that affect cuticle formation can increase permeability and so render *C. elegans* more susceptible to drugs (*Loer et al., 2015*; *Partridge et al., 2008*), this uniformly affects the epidermal innate immune response (*Dodd et al., 2018*; *Zugasti et al., 2016*). We therefore used RNAi to interfere directly with tubulin gene expression. Since eliminating MTs provokes developmental delay and lethality, worms were subjected to a relatively brief (24 hr) RNAi treatment from the L4 stage. Under these conditions, worms completed their development normally and exhibited no overt morphological defects. We chose one tubulin alpha *tba-2* and one tubulin beta *tbb-2*, as these genes have been reported to be highly expressed in the epidermis (*Cao et al., 2017*; *Hutter and Suh, 2016*). In young adult *tba-2*(RNAi) or *tbb-2*(RNAi) worms, EB1::GFP comets were no longer visible suggesting that MT dynamics had been severely compromised (*Figure 4A and B* and *Figure 4—figure supplement 1A*). Interestingly, this effect was specific to the epidermis (hyp7) since highly motile comets were still visible in the seam cells (*Figure 4A and B*). Upon wounding, EB1::GFP was not recruited to the wound site (*Figure 4A and C*). This suggests that constant expression of *tba-2* and *tbb-2* genes is required for the normal dynamic behaviour of the non-centrosomal MTs in the epidermal syncytium, as well as for the redistribution of EB1 caused by injury.

## Abrogation of MT dynamics decreases actin recruitment at the wound site

To address the question of whether MTs are required for the formation of the actin ring, we used the *tbb-2*(RNAi) conditions defined above. In resting conditions, short abrogation of *tbb-2* from the L4 stage, altered the MT network in young adult worms, as visualised with TBB-2::GFP; it became less dense and less longitudinally structured in the lateral epidermis (*Figure 4—figure supplement 1B*). The actin network was also less dense in these worms (*Figure 4—figure supplement 1B*). Upon wounding, less actin was recruited around the site of injury following *tbb-2* RNAi (*Figure 4D and G*). Further, analyses of the dynamics of radial intensities across the wound revealed that as well as accumulating to lower levels, actin was recruited later in *tbb-2* RNAi treated than in control worms (*Figure 4E–F*). Moreover, the actin ring did not constrict as quickly as in control worms, as measured by the area inside the ring after 15 min (*Figure 4H*). Notably, the relationship between MTs and actin was not reciprocal. Thus, reduction of the level of actin via inactivation of *act-2* did not change the overall pattern of MTs nor EB1 dynamics (*Figure 4—figure supplement 1A and B*). This suggests that MT dynamics do indeed facilitate actin ring closure.

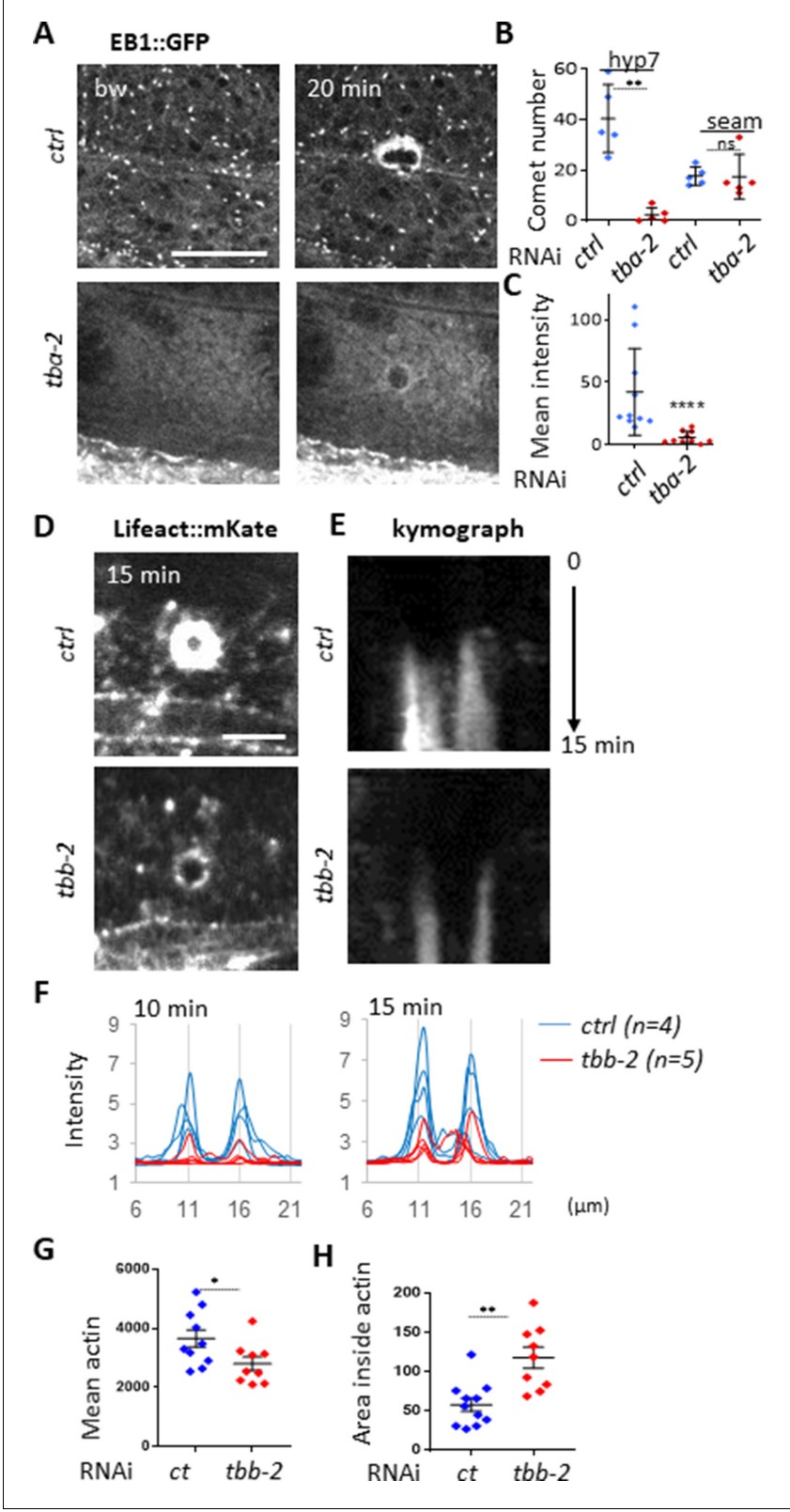

**Figure 4.** Non-centrosomal microtubule dynamics are required for actin ring closure upon wounding. (A) Tubulin α TBA-2 is required for the presence of EB1 comets and EB1 recruitment at the wound site. Representative spinning disk images of *col-19p::EBP-2::GFP* in control (*ctrl*) and *tba-2* RNAi-treated worms; time post wound [min: s]; scale bar 5 μm. (B) Quantification of EB1 comet number in either lateral epidermis or seam cells (n = 5) before *Figure 4 continued on next page*

*Figure 4 continued*

wounding. (**C**) Quantification of EB1 recruitment at the wound site, at 10 min post-wounding (n = 10). Representative spinning disk images of Lifeact::mKate (**D**) kymograph analysis (**E**), intensity across the wound at 10 and 15 min (**F**) and quantification of the intensity of actin (**G**) and the area inside the actin ring (**H**) 15 min after wounding, in control and tubulin β encoding gene *tbb-2* RNAi treated worms; ns p>0,05, *p<0.05, **p<0.01 and ****p<0.0001 non-parametric Mann-Whitney test.

The online version of this article includes the following video and figure supplement(s) for figure 4:

**Figure supplement 1.** Tubulin isotypes required for non-centrosomal MT organisation in the lateral epidermis.
**Figure 4—video 1.** The MT + tip protein dynactin DNC-2 is recruited at the wound site.
https://elifesciences.org/articles/45047#fig4video1

## Microtubule dynamics is required for the immune response

We had previously demonstrated in a genome-wide RNAi screen that knocking down certain genes associated with MT function, including *tba-2* and *tba-4*, leads to an abrogation of the induction of the AMP gene reporter *nlp-29p::GFP* usually caused by *Drechmeria coniospora* infection (*Supplementary file 1*-Table S1, *Zugasti et al., 2016*). Using the same protocol of short inactivation from the L4 stage presented above, we confirmed that inactivation of these two tubulin α genes led to a block of AMP induction after infection in the adult (*Figure 5A*). In addition, we showed that *tba-2* and *tba-4* are also required for AMP reporter gene induction after wounding (*Figure 5A*). MTs are formed by dimers of tubulin, each consisting of two subunits, tubulin α and tubulin β. In the *C. elegans* genome, there are 9 α (TBA) and 6 β (TBB) genes. Some have been shown to act in specific cells or tissue, such as *mec-7* (β) and *mec-12* (α) in the mechanosensory cells (*Savage et al., 1989*), other are predicted to be ubiquitously expressed, and several to be expressed in the epidermis (*Harris et al., 2010*). We extended our investigation to all tubulin genes by assaying the effect of individually knocking down their expression by RNAi specifically in the adult epidermis, using an *rde-1* RNAi resistant mutant rescued in the adult epidermis (*Xu and Chisholm, 2011*; *Zugasti et al., 2014*). In addition to *tba-2* and *tba-4*, we found that inactivation of tubulin β genes *tbb-1* and *tbb-2* was also associated with a block of AMP reporter gene induction upon infection (*Figure 5B and C*). We found that this was not a consequence of reduced spore binding to the worm cuticle (*Figure 5—figure supplement 1A*). Using the TBB-2::GFP strain, we also showed that inactivation of the genes that blocked the immune response altered the pattern of MTs, confirming the expression and functional importance of these tubulin isotypes in the adult epidermis (*Figure 5—figure supplement 2*).

On the other hand, inactivation of tubulin-γ (TBG-1) expression in the adult epidermis a few hours before infection did not affect AMP reporter gene induction nor the overall MTs pattern (*Figure 5C* and *Figure 5—figure supplement 2*). Since tubulin-γ nucleates MTs at their minus end (*Quintin et al., 2016*; *Wang et al., 2015*), this could suggest that unlike MT + ends, – ends are less dynamic in the adult epidermis and are not required for the immune response upon wounding. As previously shown in the *C. elegans* epidermis (*Wang et al., 2015*), however, TBG-1 could act redundantly with the minus-end stabiliser PTRN-1/patronin, which has a known role in epidermal wound repair (*Chuang et al., 2016*). Thus, independent of their role in the maintenance of epidermal structure, and in addition to a function in mediating wound closure upon wounding, MT + end dynamics appear to be required for the regulation of the transcriptional response to injury and fungal infection in the epidermis.

## SNF-12, RAB-5 and RAB-11 get locally recruited upon wounding

We hypothesised that microtubules could be required for controlling the subcellular localization of key signalling molecules that are necessary for the induction of the epidermal immune response. The SLC6 family member SNF-12 was a prime candidate, as we have previously shown that it is localized in clusters at the apical surface of the hyp7 syncytium (*Dierking et al., 2011*). We therefore characterised further the localization of SNF-12 and its behaviour following wounding. We performed colocalisation analyses between SNF-12::GFP and available vesicular and plasma membrane markers (early, late and recycling endosomes, lysosomes, plasma membrane lipids). We did not detect any colocalisation, but we observed that SNF-12 clusters were in the same focal plane as $PIP_2$ and the most apical RAB-11 recycling endosomes, and apical to LAAT-1-positive lysosomes (*Figure 6A and*

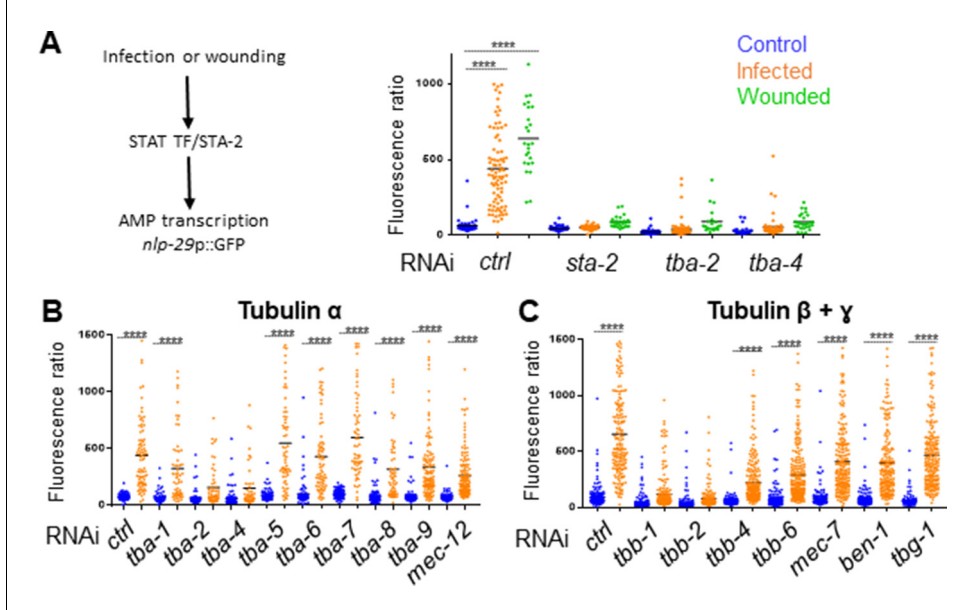

**Figure 5.** Specific tubulin isoforms are required for the activation of the immune response upon wounding and fungal infection. Quantification of green fluorescence in a strain carrying the *nlp-29p::GFP* transcriptional reporter after RNAi against different tubulin α, β and γ genes in non-infected worms or after infection with *D. coniospora* or after wounding (blue, orange and green symbols, respectively). The ratio between GFP intensity and size (time of flight; TOF) is represented in arbitrary units. (**A**) Worms were fed on RNAi bacteria from the L4 stage and after 24 hr infected or wounded; *sta-2*(RNAi) is known to block the immune response (*Dierking et al., 2011*). (**B, C**) Worms sensitive to RNAi primarily in the adult epidermis were fed on RNAi clones from the L1 stage and infected with *D. coniospora* at the young adult stage. Mean are represented in black, numbers of worms in 5A: 47, 87, 25, 48, 46, 26, 66, 107, 21, 33, 52, 28; in 5B: 106, 82, 67, 59, 63, 57, 112, 51, 95, 65, 88, 76, 79, 65, 128, 58, 98, 121, 69, 138; in 5C: 228, 180, 199, 210, 201, 162, 145, 232, 220, 224, 147, 207, 196, 188, 133, 209. Only ****p<0.0001 is presented; ANOVA Bonferoni's test. Graphs are representative of the results obtained from at least three independent replicates.

The online version of this article includes the following figure supplement(s) for figure 5:

**Figure supplement 1.** Worms are efficiently infected after knock-down of tubulin gene expression by RNAi.

**Figure supplement 2.** RNAi against *tba-2*, *tba-4*, *tbb-1* and *tbb-2*, but not *tbg-1* alters the pattern of MTs in the adult epidermis.

---

*B*, *Figure 6—figure supplements 1* and *2*). This indicates that SNF-12 is in a yet-to-be defined apical membrane compartment. Interestingly, in the dorsal and ventral epidermis, SNF-12 was found in a banded pattern, like the highly organised circumferential cytoskeleton (*Figure 6C*, *Figure 6—video 1*). Compared to EB1 and RAB-11 (see below), the majority of SNF-12 clusters were static or just vibrating, with only very few moving longer straight distances, presumably along MT tracks, with a low speed of 0.017 ± 0.005 μm/s (*Figure 6D*, *Figure 6—video 1*, *Figure 2—source data 1*).

When we wounded worms carrying the SNF-12::GFP reporter, we observed a progressive recruitment of SNF-12 to the wound site (*Figure 6E and F*, *Figure 6—video 2*), with the majority of the clusters around the wound moving in a directed manner at a low speed of 0.007 ± 0.004 μm/s (*Figure 2—source data 1*). A similar recruitment was also seen in several other strains containing different tagged forms of SNF-12 (*Figure 6—figure supplement 3*) suggesting that it reflects the behaviour of the endogenous protein. When we disrupted MTs, via RNAi of *tba-2*, SNF-12 patterning and recruitment were severely compromised (*Figure 6G–I*). Similarly, both EB1 and SNF-12 patterning and dynamics upon wounding were also affected when the MT severing protein SPAS-1 was overexpressed in the worm epidermis (*Figure 6—figure supplement 4*). Together, these results suggest that MTs play an important role in SNF-12 localisation and dynamics and thereby in the induction of AMP gene expression.

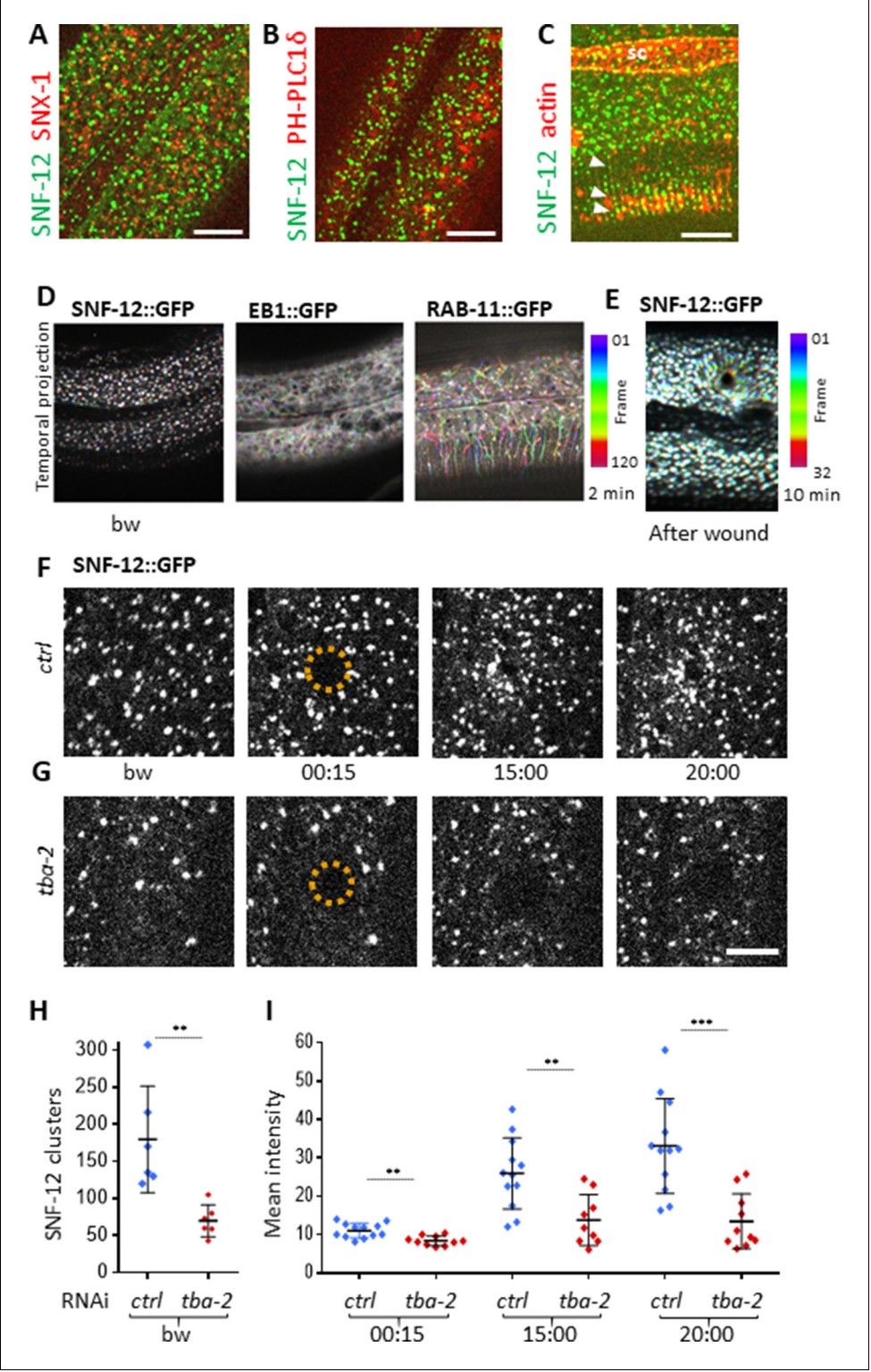

**Figure 6.** SNF-12 localizes to apical clusters that are recruited at the wound site in a MT-dependent way. Representative spinning disk images of worms carrying *col-12p::SNF-12::GFP* as well as a red marker of early endosome (*snx-1p::mRFP::SNX-1*; **A**), membrane lipids (PIP₂;*ced-1p::mCherry::PH-PLC1δ*; **B**) or actin (*col-62p:: Lifeact::mKate*; **C**); scale bar 10 μm. (**D**) The different dynamics of SNF-12, EB1 and RAB-11 are represented with a temporal color-coded projection of 120 frames over 2 min (one fps) before wounding. (**E**) After wounding, SNF-12::GFP clusters move towards the wound as represented with a color coded temporal projection of 32 frames

*Figure 6 continued on next page*

*Figure 6 continued*

over 10 min before wounding and 40 after, 6.30 min after wounding. (F–G) The SNF-12 recruitment to the wound site seen in control animals (F) is abrogated upon *tba-2* RNAi (G). The dashed circle is centred on the wound site; time post wound [min:s]; scale bar 5 μm. (H) Quantification of SNF-12 cluster number in the lateral epidermis (worm n = 6). (I) Quantification of SNF-12 recruitment at the wound site at different time points post wounding (wound n = 12 for control worms and 10 for *tba-2* RNAi). **p<0.01 and ***p<0.001; non-parametric Mann-Whitney test.

The online version of this article includes the following video and figure supplement(s) for figure 6:

**Figure supplement 1.** SNF-12 does not colocalise with RAB-11 or LAAT-1.

**Figure supplement 2.** SNF-12 exhibits a specific dynamic behaviour upon wounding.

**Figure supplement 3.** SNF-12 recruitment towards the wound site is observed using different chimeric reporter proteins.

**Figure supplement 4.** Spastin-1 overexpression drastically alters MT and SNF-12 dynamics before and after wounding.

**Figure 6—video 1.** SNF-12 dynamics in adult epidermis.

https://elifesciences.org/articles/45047#fig6video1

**Figure 6—video 2.** SNF-12 vesicles are recruited at the wound site.

https://elifesciences.org/articles/45047#fig6video2

**Figure 6—video 3.** RAB-5 dynamics in adult epidermis.

https://elifesciences.org/articles/45047#fig6video3

**Figure 6—video 4.** RAB-5 recruitment at the wound site.

https://elifesciences.org/articles/45047#fig6video4

**Figure 6—video 5.** RAB-11 dynamics in adult epidermis.

https://elifesciences.org/articles/45047#fig6video5

**Figure 6—video 6.** RAB-11 recruitment at the wound site.

https://elifesciences.org/articles/45047#fig6video6

The behaviour of SNF-12 was unusual as in almost all cases when we wounded strains in which vesicles markers were fluorescently tagged, there was no such recruitment (*Figure 6—figure supplement 2*). Thus, the movement of SNF-12 towards the wound site was not simply a reflection of cytoplasmic flow. Two exceptions were RAB-5, a marker of early endosomes, and RAB-11. RAB-5::GFP appeared localised in large static donut-shaped structures (0.3–0.8 μm) arranged in a reticular pattern, exchanging with smaller vesicles (0.2–0.4 μm), as described previously (*Chuang et al., 2016*; *Figure 7A*, *Figure 6—video 3*). Upon wounding these small vesicles accumulated around the wound site within 10–15 min (*Figure 7A*, *Figure 6—video 4*). RAB-11::GFP was found in extremely motile (1.31 ± 0.33 μm/s) vesicles (0.2–0.4 μm) (*Figure 2—source data 1*). They moved long distances in one direction, presumably on microtubule tracks (*Figure 6D*, *Figure 6—video 5*). Upon wounding they started to accumulate around the wound site within 10–15 min (*Figure 7B*, *Figure 6—video 6*) and reached a maximum density between 30 and 60 min post wounding. Notably, we previously demonstrated that both *rab-5* and *rab-11* are required non-redundantly for the epidermal innate immune response to infection (*Dierking et al., 2011*; *Zugasti et al., 2016*). Together, these results lead us to propose a timeline of events after wounding (*Figure 8*, *Figure 8—figure supplement 1*). In between the reorganisation of PIP$_2$ membrane subdomains and the restoration of membrane integrity involving the endocytic pathway, an actin ring forms in a myosin-independent, ARP2/3-dependent manner, tightly linked to the recruitment of MT + tip binding-protein EB1. The coordinated recruitment of endosomal vesicles and the uncharacterised SNF-12-associated clusters, which are both dependent upon MT reorganisation, tie together wound repair and the innate immune response.

## Discussion

Previous studies of wound healing in *C. elegans* used either imprecise and error prone mechanical injury with a needle, or sophisticated laser rigs not accessible to many laboratories. We demonstrated that a simple FRAP laser can be used to induce wounds with high precision, and can be combined easily with fast image acquisition. Using this system, we observed in the *C. elegans* epidermis the phenomenon of rapid membrane reorganisation previously described in *Xenopus*

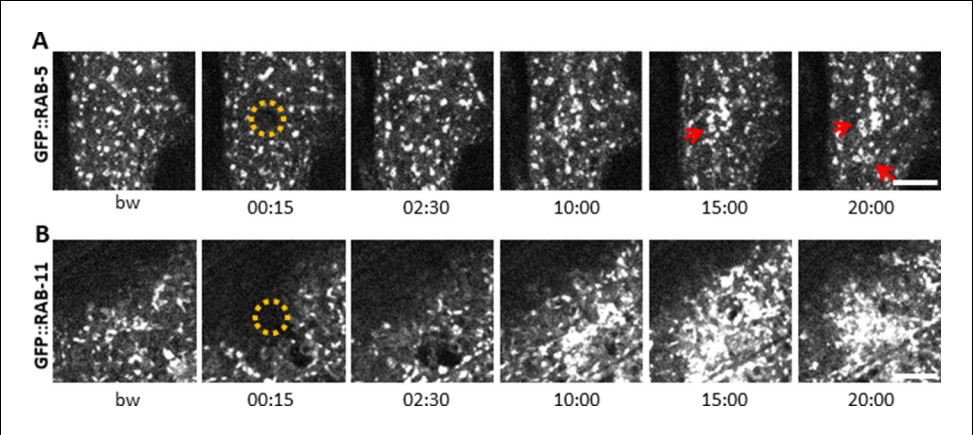

**Figure 7.** Early and recycling endosomes are recruited to the wound site. Representative spinning disk images of worms carrying *dpy-7p::GFP::RAB-5* (**A**) and *dpy-7p::GFP::RAB-11* (**B**). A fraction of the early endosome marker RAB-5 localizes to donut-shaped structures (red arrows). Upon laser wounding these RAB-5-positive structures (**A**) as well as RAB-11-positive recycling endosomes (**B**) are recruited to the wound site within 10 min. The dashed circle is centred on the wound site; time post wound [min:s]; scale bar 5 μm.

(*Davenport et al., 2016*), suggesting that these initial steps of plasma membrane repair, as well as subsequent steps in wound healing (*Nakamura et al., 2018*), may be conserved throughout Metazoans.

In addition to the acute patching of the membrane, wounding was followed by reorganisation of $PIP_2$ domains. $PIP_2$ is known to influence the adhesion between the actin-based cortical cytoskeleton and the plasma membrane (*Raucher et al., 2000*). Thus, its redistribution could be linked to the dramatic changes in the actin cytoskeleton that occurred in the minutes after wounding. The *C. elegans* epidermis is bounded on its apical side by a flexible collagen-rich cuticle that serves as an exoskeleton. As in plant cells, this is likely to give the structural support that is otherwise provided by a highly organised cortical actin network in most animal cells (*Salbreux et al., 2012*). This may explain why cortical actin is not organized in the adult lateral epidermis. During moulting, however, when the cuticle is remodelled, a highly organized cytoskeleton comprised of circumferential actin and microtubule bundles forms transiently, disappearing when the new cuticle is formed. This parallels the situation in the epidermis upon injury. Actin is rapidly assembled around the wound, and disassembled once it is repaired. Thus changes in membrane tension, either linked to alterations of the cuticle during development, or caused by a breach in its integrity are closely linked to cytoskeleton dynamics.

Wound repair has been previously suggested to involve a recapitulation of developmental programs (*Martin and Parkhurst, 2004*; *Razzell et al., 2014*). Thus for example, in the syncytial *Drosophila* early embryo, wound closure occurs through a myosin-II-dependent purse-string mechanism, driven mainly by the contractile force exerted by myosin (*Abreu-Blanco et al., 2011*; *Bement et al., 1999*). This parallels the tight control exerted by myosin on the actin cytoskeleton during morphogenetic cell shape changes in *Drosophila* embryogenesis (*Lecuit and Lenne, 2007*). In *Xenopus* oocytes, if myosin motor activity is inhibited, wounds still close, though at a much slower rate. The residual force driving wound closure is thought to be given by the treadmilling of actin, with a preferential recruitment of Rho GTPase at the wound (*Burkel et al., 2012*). The recruitment we observed of ARX-2/ARP2 to wounds reinforces the hypothesis that the actin ring in the *C. elegans* epidermis closes due to polymerisation of branched actin networks.

Actin-dependent processes, including cell migration and cell protrusion, can be influenced by MT dynamics. The mechanisms that underlie their functional crosstalk are still being characterised (*Dogterom and Koenderink, 2019*). MT + tip proteins like EB1 have been suggested to be involved in microtubule guidance, attachment to cellular structure or acting as concentration devices for signalling events (*Akhmanova and Steinmetz, 2015*). For example, a recent in vitro study showed that the MT + end-associated protein CLIP-170, together with the formin mDia1, can accelerate the actin filament polymerization which occurs at the tip of growing MTs (*Henty-Ridilla et al., 2016*). We showed here that in *C. elegans* the recruitment of MT + end binding protein EB1 to the wound is

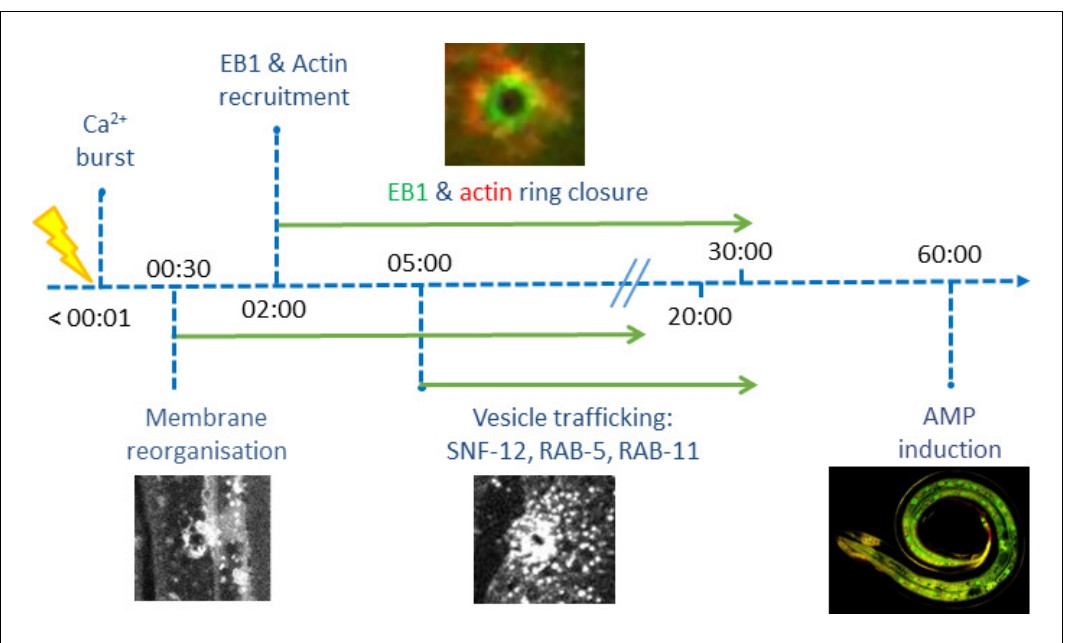

**Figure 8.** Time line of events.

The online version of this article includes the following figure supplement(s) for figure 8:

**Figure supplement 1.** Onset of events after wounding the lateral epidermis.

concomitant with that of actin. The two proteins then converge during wound closure at the same rate, with EB1 being at the leading edge. Both in vitro and in vivo studies have indicated that EB1 can interact directly with actin (*Alberico et al., 2013*). We demonstrated that perturbation of MT dynamics in vivo slows actin ring closure in *C. elegans* epidermis. Together, this suggests that MT dynamics could facilitate actin ring closure by bringing EB1 to the wound site. This MT-dependent mechanism, which is independent of myosin-based contractility (*Xu and Chisholm, 2011*), could arise because the epidermis is bounded by a rigid apical extracellular matrix, the tensile cuticle exoskeleton.

Injuring the epidermis not only leads to wound healing, but also provokes an innate immune response. Previous results in *C. elegans* suggested that the pathways involved in these two processes were distinct. Thus, although mutants for the p38 MAPK PMK-1 and for the SARM orthologue TIR-1, that acts upstream of the p38 cascade, are defective for the induction of AMP gene expression after wounding (*Pujol et al., 2008a*), they exhibit normal actin ring closure (*Xu and Chisholm, 2011*). If innate immune signal transduction pathway components are not required for wound repair, our results suggest, on the other hand, that cytoskeleton components can play a role in regulating AMP production. Indeed, several tubulin genes are required for AMP expression upon infection and wounding, potentially through their effect on the localization and dynamics of innate immune signalling proteins including SNF-12. The fact that overexpression in the epidermis of the MT severing protein Spastin caused a similar effect on SNF-12 dynamics as knocking down tubulin expression, supports a role for MTs in positioning SNF-12-containing vesicles. Determining the precise nature of these vesicles, their connection with MTs and the mode of SNF-12 activation remain challenges for the future. We speculate that this spatial regulation of signalling may contribute to restricting defence gene activation to the immediate vicinity of a wound or site of infection within the major epidermal syncytium.

A further link between cytoskeleton reorganization and AMP production comes from the study of the protein DAPK-1, a negative regulator of both wound closure and innate immune gene expression (*Tong et al., 2009*). Interestingly, *dapk-1* mutants have defects in the epidermis that were shown to result from the hyperstabilisation of MTs. DAPK-1 inhibits the function of PTRN-1, a MT-end binding protein, which acts as a MT nucleator and promotes MT stabilization. Loss of *ptrn-1* function suppresses the epidermal defects seen in *dapk-1* mutants as well as their constitutive AMP

gene expression and accelerated wound closure. Loss of *dapk-1* function is associated with an increase in EB1 comets. This too is suppressed in a *dapk-1;ptrn-1* double mutant (*Chuang et al., 2016*). Taken together with our results, this suggests that the accelerated wound healing seen in *dapk-1* mutant worms may be due to an increased recruitment of EB1 protein thus speeding up MT-dependent actin ring closure.

The cytoskeleton is an assembly of co-regulating components that function together in a precise and dynamic way. Our results suggest that this co-regulation extends to the coordination of physical changes required for membrane wound healing and immune signal transduction driving transcriptional responses to injury.

## Materials and methods

### Nematode strains

All *C. elegans* strains were maintained on nematode growth medium (NGM) and fed with *E. coli* OP50, as described (*Stiernagle, 2006*): the wild-type N2, IG274 *frIs7[col-19p::DsRed, nlp-29p::GFP]* IV (*Pujol et al., 2008a*), IG823 *frIs43[col-12p::SNF-12::GFP, ttx-3p::DsRed2]* and IG1235 *cdIs73[RME-8::mRFP, ttx-3p::GFP, unc-119(+)]; frIs43[col-12p::SNF-12::GFP, ttx-3p::DsRed2]* (*Dierking et al., 2011*), IG1327 *rde-1(ne219) V; juIs346[col-19p::RDE-1, ttx-3p::GFP]* III; *frIs7[nlp-29p::GFP, col-12p::DsRed]* IV (*Zugasti et al., 2014*), CZ13896 *juIs319[col-19p::GCaMP3, col-19p::tdTomato]* (*Xu and Chisholm, 2011*), CZ14453 *juEx3762[col-19p::EBP-2::GFP, ttx-3p::RFP]*, JLF302 *ebp-2(wow47[EBP-2::GFP]) II; zif-1(gk117) III* (*Sallee et al., 2018*), CZ14748 *juIs352[GFP::moesin, ttx-3p::RFP]* I, CZ21789 *juSi239[col-19p::GFP::TBB-2]* I, CZ9334 *juEx1919[dpy-7p::GFP::RAB-5, ttx-3p::RFP]* (*Chuang et al., 2016*), SA854 *tbb-2(tj26[GFP::TBB-2])* III (*Honda et al., 2017*), SV1009 *Is[wrt-2p::GFP::PH-PLC1δ, wrt-2p::GFP::H2B, lin-48p::mCherry]* (*Wildwater et al., 2011*), BOX188 *maph-1.1 (mib12[GFP::maph-1.1]) I* (*Waaijers et al., 2016*), GCP417 *dnc-2[prt42(3xflag::GFP::DHC-2)] III* (*Barbosa et al., 2017*), GN675 *tba-1(pg77[TagRFP-T::TBA-1]) I* (*Lockhead et al., 2016*), GOU2047 *arx-2(cas607[GFP::ARX-2])* (*Wu et al., 2017*), MBA365 *Ex[dpy-7p::GFP::CAAX, myo-2p::GFP]* kindly provided by M. Barkoulas (UCL), RT343 *pwIs82[snx-1p::mRFP::SNX-1, unc-119(+)]* (*Sato et al., 2014*, NP878 *cdIs73[RME-8::mRFP, ttx-3p::GFP, unc-119(+)]* (*Shi et al., 2009*), XW10992 *qxIs513[ced-1p::mCherry::PLC-1-PH]* and XW9653 *qxIs68[ced-1p::mCherry::RAB-7]* (*Liu et al., 2012*), ML1896 *mcIs35 [lin-26p::GFP::TBA-2, pat-4p::CFP, rol-6(su1006)]; mcIs54[dpy-7p::SPAS-1_IRES_NLSmCherry, unc-119(+)]* X (*Wang et al., 2015*). All the multiple reporter strains generated in this study were obtained by conventional crosses (see *Supplementary file 2*-Key resource table for a list of all strains).

### Constructs and transgenic lines

*frSi9* is a single copy insertion on chromosome II at the location of the Nemagenetag Mos1 insertion (*Vallin et al., 2012*) *ttTi5605* of pNP151 (*col-62p::Lifeact::mKate2_c-nmy3'utr*). pNP151 was obtained by insertion of the *col-62* promoter fused to Lifeact::mKate2 (*Reymann et al., 2016*) into the MosSCI vector pCFJ151 (*Frøkjaer-Jensen et al., 2008*). It was injected into the EG6699 strain at 20 ng/µl together with pCFJ90 (*myo-2p::mCherry*) at 1.25 ng/µl, pCFJ104 (*myo-3p::mCherry*) at 5 ng/µl, pMA122 (*hsp16.41p::PEEL-1*) at 10 ng/µl, pCFJ601 (*eft-3p::Mos1 transposase*) at 20 ng/µl and pNP21 (*unc-53pB::GFP Stringham et al., 2002*) at 40 ng/µl. A strain containing the insertion was obtained following standard selection and PCR confirmation (*Frøkjaer-Jensen et al., 2008*).

*frSi13* is a single copy insertion on chromosome II (*ttTi5605* location) of pNP159 (*dpy-7p::GFP::RAB-11*) by CRISPR using a self-excising cassette (SEC) (*Dickinson et al., 2015*). pNP159 was obtained by insertion of *dpy-7p::GFP::RAB-11* (kindly provided by Grégoire Michaux) into the pNP154 vector. pNP154 was made from a vector containing the SEC for single insertion on Chromosome II at the position of *ttTi5605* (pAP087, kindly provided by Ari Pani). pNP159 was injected in N2 at 10 ng/µl together with pDD122 (*eft-3p::Cas9*) at 40 ng/µl, pCFJ90 (*myo-2p::mCherry*) at 2.5 ng/µl, pCFJ104 (*myo-3p::mCherry*) at 5 ng/µl, and #46168 (*eef-1A.1p::CAS9-SV40_NLS::3'tbb-2 Friedland et al., 2013*) at 30 ng/µl. Non-fluorescent roller worms were selected then heat shocked to remove the SEC by FloxP as described in *Dickinson et al. (2015)*. All constructs were made using Gibson Assembly (NEB Inc, MA) and confirmed by PCR or sequencing. Plasmid sequences are available upon request.

Different transgenic strains containing several tagged version of SNF-12 were generated including IG1784 *frEx597[pSO16(col-12p::SNF-12::mKate_3'unc-54), pCFJ90(myo-2p::mCherry]*, IG1270 *frEx453[pMS8(col-12p::GFP::STA-2), pMS9(col-12p::mCherry::SNF-12)]* and IG1663 *frEx577[pNP158 (snf-12p::SNF-12::GFP_3'snf-12, pCFJ90(myo-2p::mCherry), pCFJ104(myo-3p::mCherry)]*.

## RNAi

The RNAi bacterial clones were obtained from the Ahringer library (*Kamath et al., 2003*) or Vidal library (*Rual et al., 2004*) and sequenced to confirm their identity using CloneMapper (*Thakur et al., 2014*). RNAi feeding experiments were performed at 25°C and the RNAi clone *sta-1* was used as a negative control. To avoid developmental delay or lethality, RNAi experiments of tubulin or actin genes were either performed from the L4 stage or in a strain allowing gene silencing primarily in the epidermis from the L4 stage (IG1327 (*rde-1(ne219) V; juIs346[col-19p::RDE-1, ttx-3p::GFP] III; frIs7 [nlp-29p::GFP, col-12p::DsRed] IV*) *Zugasti et al., 2016*), by rescuing the *rde-1* RNAi resistant mutant in the adult epidermis as previously described (*Xu and Chisholm, 2011*).

## Infection and needle wound

Eggs prepared by the standard bleach method were allowed to hatch in 50 mM NaCl in the absence of food at 25°C over night. Synchronized L1 larvae were transferred to NGM agar plates spread with *E. coli* OP50 and cultured at 25°C until the L4 stage (40 hr) before being exposed to fungal spores as previously described (*Pujol et al., 2001*). To count the number of spores, young adult worms were infected for 8 hr at 25°C then observed on a slide under the microscope. Statistical significance was determined using ANOVA Bonferoni's test (Graphpad Prism). Needle wounding was performed as previously described (*Pujol et al., 2008a*) with a standard microinjection needle under a dissecting microscope by pricking the worm's posterior body or tail on agar plates; worms were analysed after 6 hr.

## Fluorescent reporter analyses

Analysis of *nlp-29*p::GFP expression was quantified with the COPAS Biosort (Union Biometrica; Holliston, MA) as described in *Pujol et al. (2008b)*. In each case, the results are representative of at least three independent experiments with more than 70 worms analysed. The ratio between GFP intensity and size (time of flight; TOF) is represented in arbitrary units. Statistical significance was determined using a non-parametric analysis of variance with a Dunn's test (Graphpad Prism). Fluorescent images were taken of transgenic worms mounted on a 2% agarose pad on a glass slide anesthetized with 0.01% levamisole, using the Zeiss AxioCam HR digital colour camera and AxioVision Rel. 4.6 software (Carl Zeiss AG).

## Image acquisition and laser wound

Since in preliminary experiments we observed that immobilization using latex beads impacted vesicle and protein dynamics in the epidermis, young adult worms (containing less than five eggs) were immobilized on a slide using 0.01% levamisole. Time lapse and/or Z stack were acquired using two different spinning disk microscopes: an inverted Visitron Systems GmbH spinning disk with a Nikon 40X oil 1.3 NA objective and 1.5X lens, and a Roper inverted spinning disk with a Nikon 100X oil 1.4 NA objective, were used with their respective acquisition software (VisiView or Metamorph). For spatially resolved time-lapse imaging, a Zeiss LSM780 confocal microscope whose GaAsP detector was chosen for its high sensitivity and controlled through the Zen acquisition software.

Laser wounding was performed by using a 405 nm laser at its maximum power using either Visitron, iLas FRAP or Zeiss FRAP systems. The respective laser power before objectives was: 7.3 mW, 1–2 mW and 3 mW. 1–3 s of laser pulses were applied on a circular region of interest (ROI) of 8 × 8 pixels with the Visitron (~2.13 μm in diameter), 21 × 21 pixels with the Nikon (~1.68 μm in diameter) or with the confocal (~1.8 μm in diameter) on the apical plane of the lateral epidermis either left or right of the seam cells. Each worm was wounded from 1 to 3 times.

Time lapse acquisitions were 10 to 20 min long. Z stack acquisitions were performed on 4–6 planes spaced by 0.3 or 0.5 μm. High temporal resolution was obtained on one single plane with a time interval of 0.2 to 0.5 ms. Laser wound was performed between the first and the second Z stack. For colocalisation analysis, we initially controlled that there was no signal cross talk using single

labelled strains. We then performed sequential acquisitions on the strains expressing two fluorescent proteins. Experiments requiring comparative analysis were performed on the same day.

## Image analysis

All image processings were done using Fiji software. A custom Image J macro was used to correct for Z-drift, while XY drift was corrected either with the Image Stabilizer or the Correct 3D drift plugin from ImageJ or the online available macro 'NMSchneider/FixTranslation'.

To analyse EB1 density before wounding, we manually counted fluorescent dots as described in *Chuang et al. (2016)*, in a 300 $\mu m^2$ ROI in the lateral hyp7 or in an 80 $\mu m^2$ ROI in the seam cell in five worms per RNAi condition. Similarly, we measured SNF-12 cluster density in the lateral hyp7, analysing six worms per RNAi condition. Quantification of EB1 recruitment at the wound site was performed by measuring the RawIntDen (sum of the pixel intensities) of a selected ROI of 256 $\mu m^2$ for EB1 centred at the wound site. Since EB1 brightness differed before wounding between worms, for each single wound, only pixels with values above a defined brightness threshold were considered, the threshold being set to the [mean + 3 x standard deviation] in the ROI before wounding. Quantification of SNF-12 recruitment was performed by measuring the mean intensity of a selected ROI of 18 $\mu m^2$ centred at the wound site at specific time points. Statistical significance was determined using a non-parametric Mann-Whitney test (Graphpad Prism).

To visualise the speed and directionality of EB1, RAB-11 and SNF-12, a temporal projection was done using a temporal-Color code (plugin from Kota Miura) on 120 frames over 2 min (one frame per second) before wounding for all tree makers and on 32 frames over 10 min after wounding for SNF-12.

To analyse particle motility before and after wounding, we used the macro KymographClear described in *Mangeol et al. (2016)*. For EB1, RAB-11 and SNF-12, we acquired movies with time interval of 300 ms, 400 ms and 200 ms, respectively. Since SNF-12 was observed to move much slower than the other dynamic proteins studied, a substack with a time interval of 20 s was generated. SNF-12 quantifications were done for particles with a minimum displacement of 5 pixels within 120 s for before wounding and within either 210 or 420 s for after wounding. To analyse the directionality of EB1 movement, we developed a workflow using first available Image J/Fuji plugins to correct image drift ('Image stabilizer'), crop movies to a standard size and number of frames (100*100 pixels *1000;"crop' and 'make substack'), subtract the stable background with a median temporal filter ('median 3D', treating the time series as a Z-stack, setting the filter at 31 (temporal), and then subtracting from each original image). We then defined particles using the Trackmate plugin (*Tinevez et al., 2017*), with estimated blob diameter of 3 pixels, and a threshold at 900, linking distance and gap-closing max distance two pixels, gap-closing max frame gap of 1. From the resulting file, the track ID and x,y positions were analysed using a custom R script that calculated vectors relative to the wound site for each moving particle.

To compare the actin ring closure upon different RNAi condition, a ROI of 80 $pixels^2$ located on the centre of the wound was selected 15 min post-wounding. The area inside the actin ring was detected using the Huang automatic threshold method from Image J. To first compare the recruitment of EB1 versus actin, the radial profile (plugin from Paul Baggethun) was calculated, after determination of the centroid of the wound for each time point by manually selecting the wound area. For each radial profile, we extracted the radius corresponding to the maximum intensity and plotted this radius versus time. To further quantify, we build kymographs by considering masks that corresponds to stripes oriented in the anterior-posterior (AP) direction and including the ablation region. We obtained one-dimensional profiles by averaging the signal intensity over the circumferential direction; to account for variations in signal intensity during time, we normalised each profile by its maximal value. These colour-coded profiles were concatenated to produce a kymograph.

To analyse the microtubule orientation, we obtained maps of the local orientation of microtubules of an ROI of 20 × 20 $\mu m$ of confocal images of MAPH-1::GFP reporter strain using the ImageJ plugin OrientationJ and custom Matlab scripts (Source file plotorientation), which provide the direction of the largest eigenvector of a local intensity structure tensor (*Rezakhaniha et al., 2012*). We then estimated the angle θ between the local direction of microtubule bundle and the direction to a point of reference (the centre of the wound region); θ lies in the 0 to +90 sector since it measures the discrepancy between two directions. We then measured the averaged angle θ over three annular sectors labelled according to their distance to the centre of the wound: R1, R2, R3. Statistics were

performed in Matlab(R) using the circular Statistic toolbox (*Berens, 2009*). Custom R and Matlab scripts are either available as source file or upon request.

## Acknowledgements

We thank Annie Bonnet, Vincent Rouger, Guillaume Bordet, Sham Tlili, Claudio Collinet and Pierre Mangeol for their contributions, Asako Sugimoto, Andrew Chisholm, Michel Labouesse, Sander van den Heuvel, Michalis Barkoulas, Bart Grant, Xiaochen Wang, Jessica Feldman, Reto Gassmann and the *Caenorhabditis* Genetics Center (University of Minnesota, Minneapolis, MN) supported by the National Institutes of Health Office of Research Infrastructure Programs (P40 OD010440) for strains, Ari Pani, Anne Cecile Rayman, Gregoire Michaux and Michael Sixt for plasmids, Chris Crocker at Wormatlas (supported by NIH OD010943 to David Hall) for diagrams, and Didier Marguet, Ariane Abrieu, Alphée Michelot, Thomas Lecuit and members of our lab for discussions and critical reading of the manuscript. Worm sorting was performed by Jerome Belougne using the facilities of the French National Functional Genomics platform, supported by the GIS IBiSA and Labex INFORM. We thank the imaging core facility (ImagImm) of the Centre d'Immunologie de Marseille-Luminy (CIML) supported by the French National Research Agency program (France-BioImaging ANR-10-INBS-04), and Centuri funded by the « Investissements d'Avenir » French Government program managed by the French National Research Agency (ANR-16-CONV-0001) and from Excellence Initiative of Aix-Marseille University - A*MIDEX.

## Additional information

### Funding

| Funder | Grant reference number | Author |
|---|---|---|
| Agence Nationale de la Recherche | ANR-16-CE15-0001-01 | Clara Taffoni<br>Sébastien Mailfert<br>Mathieu Fallet<br>Jonathan J Ewbank<br>Nathalie Pujol |
| Institut National de la Santé et de la Recherche Médicale | | Jonathan J Ewbank |
| Centre National de la Recherche Scientifique | | Shizue Omi<br>Sébastien Mailfert<br>Mathieu Fallet<br>Jean-François Rupprecht<br>Nathalie Pujol |
| Aix-Marseille Université | | Clara Taffoni<br>Caroline Huber |
| Agence Nationale de la Recherche | ANR-12-BSV3-0001-01 | Clara Taffoni<br>Sébastien Mailfert<br>Mathieu Fallet<br>Jonathan J Ewbank<br>Nathalie Pujol |
| Agence Nationale de la Recherche | ANR-11-LABX-0054 | Clara Taffoni<br>Sébastien Mailfert<br>Mathieu Fallet<br>Jonathan J Ewbank<br>Nathalie Pujol |
| Agence Nationale de la Recherche | ANR-11-IDEX-0001-02 | Clara Taffoni<br>Sébastien Mailfert<br>Mathieu Fallet<br>Jonathan J Ewbank<br>Nathalie Pujol |
| Agence Nationale de la Recherche | ANR-10-INBS-04-01 | Clara Taffoni<br>Sébastien Mailfert<br>Mathieu Fallet<br>Jonathan J Ewbank<br>Nathalie Pujol |

| Agence Nationale de la Recherche | ANR-16-CONV-0001 | Shizue Omi<br>Sébastien Mailfert<br>Mathieu Fallet<br>Jean-François Rupprecht<br>Jonathan J Ewbank<br>Nathalie Pujol |
|---|---|---|

The funders had no role in study design, data collection and interpretation, or the decision to submit the work for publication.

## Author contributions

Clara Taffoni, Formal analysis, Investigation, Visualization, Methodology, Writing - original draft; Shizue Omi, Investigation, Methodology; Caroline Huber, Investigation, Visualization; Sébastien Mailfert, Mathieu Fallet, Visualization, Methodology; Jean-François Rupprecht, Software, Validation, Methodology; Jonathan J Ewbank, Funding acquisition, Writing - original draft, Writing - review and editing; Nathalie Pujol, Conceptualization, Formal analysis, Supervision, Funding acquisition, Validation, Writing - original draft, Project administration, Writing - review and editing

## Author ORCIDs

Clara Taffoni ⓘ https://orcid.org/0000-0001-9265-5909
Shizue Omi ⓘ http://orcid.org/0000-0002-2711-2016
Caroline Huber ⓘ http://orcid.org/0000-0003-3767-2719
Sébastien Mailfert ⓘ http://orcid.org/0000-0002-0409-0432
Mathieu Fallet ⓘ https://orcid.org/0000-0001-8144-6159
Jean-François Rupprecht ⓘ http://orcid.org/0000-0001-8904-5878
Jonathan J Ewbank ⓘ https://orcid.org/0000-0002-1257-6862
Nathalie Pujol ⓘ https://orcid.org/0000-0001-8889-3197

## Decision letter and Author response

Decision letter https://doi.org/10.7554/eLife.45047.sa1
Author response https://doi.org/10.7554/eLife.45047.sa2

# Additional files

## Supplementary files

• Source code 1. Orientation of fibers analysis.

• Supplementary file 1. MT-related genes identified in a genome-wide screen for regulators of AMP gene expression from *Zugasti et al. (2016)*.

• Supplementary file 2. Key resources table.

• Transparent reporting form

## Data availability

All data generated or analysed during this study are included in the manuscript and supporting files. Source data files have been provided for all quantitative figures.

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
