## [Decision Letter]

Thank you for submitting your article "Microtubule plus-end dynamics link wound repair to the innate immune response" for consideration by *eLife*. Your article has been reviewed by three peer reviewers, one of whom is a member of our Board of Reviewing Editors, and the evaluation has been overseen Didier Stainier as the Senior Editor. The following individuals involved in review of your submission have agreed to reveal their identity: William Bement (Reviewer #2); Andrew D Chisholm (Reviewer #3).

The reviewers have discussed the reviews with one another and the Reviewing Editor has drafted this decision to help you prepare a revised submission.

Summary:

This manuscript establishes a new model for cellular wound healing, using laser induced cutting of the *C. elegans* epidermis. The authors characterize the cellular responses to wound healing and note a remarkable accumulation of the microtubule plus end binding protein Eb1 at the wound site. They further suggest that this regulates both subsequent actin accumulation and the enrichment of a transcription factor that mediates the immune response to wounding. These interesting observations link a novel microtubule-dependent wound response to a local immune response.

Essential revisions:

1) Organization of microtubules. The data on accumulation of Eb1 puncta at the wound site is very clear. However, neither the organization of their paths, nor a clear image of microtubule organization is presented. Further investigation of this will allow increased clarity of the model for microtubule function in wound healing.

2) Quantitation of data. This manuscript characterizes a new method for wounding and describes the dynamics of various cellular components during the healing process. To make it a more valuable resource for future research, increased quantitation of the data, showing both reproducibility over a number of cells, as well as showing the kinetics of various components relative to each other should be provided.

There were a number of points that were not entirely clear in the manuscript, such as how microtubules control SNF-12 localization (and further discussed in detail below). Changes to the text to clarify many of these confusions, listed in the reviews below, would strengthen the manuscript.

Reviewer #1:

This manuscript establishes a robust assay for wounding the *C. elegans* lateral epidermis. Using this assay, the authors characterize the membrane and cytoskeletal responses to wounding, which are similar to previously studies *Drosophila* and *Xenopus* models. They also show a remarkable increase in the microtubule end-binding protein Eb1, at the wound site. Functionally, they find that disrupting microtubule dynamics inhibits actin accumulation and the normal immune response. There are several really interesting observations in this paper, however, the conclusions at present could be strengthened by increased rigor throughout as different interpretations of the data exist.

1) As they are establishing a novel wounding methodology and characterizing the response, increased quantitation of the dynamics and extent of responses in Figures 1-3 are required. For all of these, it would be ideal to have (in addition to the videos shown) quantitations over several videos to see the reproducibility of the responses. This will also be helpful in laying out the temporal order of changes occurring. I find this to be especially important as the authors suggest that the microtubule response occurs before the actin accumulation, though this is not clear from the video presented.

2) Constriction of the F-actin at the wound. This is documented for a single wound in Figure 3H, but is not obvious in the videos provided. It is important to document whether this is a constriction or just a loss of F-actin over time. This is especially confusing as the Introduction states that healing hear is not driven by acto-myosin contraction, but rather by Arp2/3 mediate F-actin assembly.

3) The accumulation of tubulin at wound sites was dramatically less than Eb1. This is perhaps surprising and should be discussed. Along these lines, it was not clear whether the authors could track Eb1 puncta within the wound site due to density of puncta. Are all puncta motile and analyzable?

4) Epistasis of actin and microtubule accumulation. The authors state in the Discussion that Eb1 accumulates before actin. However, this is not clear from the videos provided. Functionally, there is a decrease in F-actin recruitment when tubulin genes are knocked down or microtubules disrupted. However, the timepoints here were late and don't address whether initial accumulation occurs but does not increase or whether it peaks and goes down, or never increases in the first place. An additional concern along these lines is that the F-actin as reported by LifeAct appears to be significantly disrupted in the unwounded tubulin mutant epidermis (S2B – to a similar extend as the Act2 mutant). This raises the caveat that it may not be a wound-induced effect, but a general effect on cytoskeletal organization.

5) Connection to the immune response. I felt that I needed more discussion of how the SNF-12 localization connected to the phenotype. Is the localization of the SNF-12 transcription factor at the wound site necessary for its transcriptional activity? How do you imagine this occurs. Are microtubules required for MAPK kinase activity or DCAR-1 signaling in any way? Authors should rule out the possibility that microtubules affect this signaling axis and that they work through localization of SNF-12. Along these lines, in the induction of AMP's in response to cuticle defects similarly dependent on tubulin genes?

6) SNF-12 clusters were significantly lower in unwounded epidermis of tubulin mutants compared to controls. One possibility is that microtubules are required for cluster formation and not localization of SNF-12.

Reviewer #2:

While not widely recognized, the response of single cells to damage is just as important as the response of tissues to damage. Not surprisingly then, a growing body of work suggests that cell damage elicits a conserved response that involves the mobilization of a variety of signaling, lipid, and cytoskeletal components. There are at least two biological consequences of this response: First, the hole in the cell is patched or otherwise closed, preventing the cell from simply lysing. Second, the damage material is cleaned up and removed, such that the wound site is eventually restored to its original state (Sonnemann and Bement, 2011). To a first approximation, these biological imperatives parallel those of damaged tissue, with clotting being responsible for plugging the hole and the action of immune system being responsible for the clean-up.

In the current paper, Taffoni et al. ask whether the single cell wound response is somehow linked to and coordinated with the immune response in the *C. elegans* embryo. To accomplish this, they develop a laser wounding system that permits them to make relatively precise wounds in the embryonic syncytial epiderms. They first show that repair of these wounds has several characteristics in common with the repair response in other damaged cell types. They then identify a striking accumulation of EB1, a protein that binds to the plus ends of growing microtubules, around wounds, leading them to conclude that wounding elicits local microtubule reorganization around wounds. They report that they can block this response by depleting microtubule subunit proteins (i.e. alpha and beta tubulin) and that this manipulation not only impairs the immediate wound response, but also impairs the innate immune response as monitored by expression of an AMP reporter gene. They further link microtubules to recruitment of SNF-12-a protein involved in the innate immune response-to wound sites. Based on these results the authors conclude that the microtubule response is an important feature of both the basic (cell healing) wound response and the innate immune response.

Several features of this paper are very appealing: the question is of fundamental importance for those who seek to understand wound repair, the authors nicely make the link between the basics of cell damage repair as it occurs in their system and in other systems, and much of the imaging clearly illustrates the points the authors are making. Perhaps most importantly, the link between the microtubule response in a single cell (albeit a very big cell) and the mobilization of the innate immune response is an important one.

However, several points need to be addressed:

1) It is very difficult to tell what, exactly, is happening with the microtubules at the wound site because the different approaches seem to be suggesting different things. That is, in some cell types microtubules accumulate around wounds in a radial array such that many microtubules are focussed toward the wounds (e.g. Mandato and Bement, 2003). Such an arrangement would be consistent with the accumulation of the EB1-GFP fluorescence around wounds (but see below). However, it is not consistent with the movement of the EB1-GFP comets, which appear to move in all directions and in the absence of any obvious bias toward the wounds. The live imaging of the microtubules themselves (Figure 3C) is also inconsistent with a radial array. As with the movement of the comets, it seems to indicate that the microtubules are randomly oriented with respect to the wound. Now, one might be tempted to imagine that the intense EB1-GFP signal that accumulates around the edge of the wound is sufficient evidence of some kind of wound-localized array of microtubules but there are two problems with this notion. First, that signal is not obviously in comet form, rather, it just appears to be a uniform slathering of GFP on the plasma membrane. Second, EB1 only associates with growing microtubules plus ends; it does not associate with stable microtubule plus ends (e.g. Akhmanova and Steinmetz, 2015). Thus, the apparently stable accumulation of EB1-GFP at the wound edge is rather mysterious. To settle this point the authors need to fix wounded embryos and stain for microtubules or find an alternative way to characterize the organization of microtubules around wounds.

2) This may reflect my ignorance, but it is not at all clear to me what links the transport of SNF-12 to wounds and the triggering of the innate wound response. Does something special happen to the SNF-12 at wound edges? I was under the impression that SNF-12 exerts its effects on transcriptional control of the response after entering the nucleus as previously reported by the authors (Dierking et al., 2011). While it is not incumbent on the authors to trace out all of the steps in the pathway that leads to an upregulation of the immune transcriptional response, it would be helpful to have a least a little bit of information on why recruitment to the wound is important.

Reviewer #3:

Taffoni et al. examine the role of microtubule dynamics in epithelial wound responses in the epidermis of the nematode *C. elegans*. Microtubule dynamics have been implicated in several cell repair processes, and were implicated in *C. elegans* skin wound repair, but had not previously been examined directly. The authors use extensive live imaging of fluorescently tagged markers to show that MT dynamics are affected by wounding and link physical wound repair to innate immune response to damage via transport of the signalling molecule SNF-12. The authors provide extensive descriptions of the dynamics of various membrane and organelle markers after injury. Overall this is a thorough and careful study that advances our understanding of the cell biology of wound repair in vivo.

---

## [Author Response]

Essential revisions:1) Organization of microtubules. The data on accumulation of Eb1 puncta at the wound site is very clear. However, neither the organization of their paths, nor a clear image of microtubule organization is presented. Further investigation of this will allow increased clarity of the model for microtubule function in wound healing.

We thank the reviewers for their insightful comments that led us to define a more precise model. We have now better characterised EB1 and microtubule dynamics upon wounding. We have generated several new strains and combined imaging with advanced analysis techniques.

Detailed analysis of EB1 revealed 2 regimes. The first was analysed by tracking EB1 comets after wounding. It matches the typical speed of microtubule + tips. This tracking also revealed that EB1 comets were not associated with any directional bias toward the wound, but seem to stop at the wound en-passant. The second regime of EB1 starts when it accumulates at the wound in the form of a ring. Kymograph analyses of both EB1 and actin signals reveal that the EB1 ring closes at the same speed as the actin ring, while also confirming our previous observation that EB1 precedes actin, EB1 being found at the leading edge of actin while the ring closes.

We have complemented these observations with more detailed examination of MT by analysing several MT-binding protein reporters, like the dynactin protein DNC-2, known to bind MT + tips in the *C. elegans* embryo (Barbosa et al., 2017), and MAPH-1, a MT associated protein MAP1, as well as more resolutive confocal analyses of the tubulin TBB-2 reporter. The observations with DNC-2 were fully consistent with what we had seen with EB1 and confirmed an accumulation of MT + tips at the wound at the same time as actin. Second, confocal analysis revealed MT growing and retracting at the wound just before and during actin accumulation. Third, quantification of MT orientation and tracking of EB1 comets, allow us to rule out a radial organisation of MT at the wound, consistent with the non-directional dynamics of EB1 comets. These data support a dynamic orthoradial organisation of the MT at the wound, and suggest that the microtubules reorganised to knit a web around the wound as the actin ring closes.

Reviewer #2 pointed out (see below), generally, “EB1 only associates with growing microtubules plus ends; it does not associate with stable microtubule plus ends (e.g. Akhmanova and Steinmetz, 2015)”. To reconcile this model with our observations that EB1 stops at the wound site and then migrates inward with the actin as the wound closes, we propose as a working model that MT-associated EB1 binds to actin at the wound and then dissociates from MT during closure. The actin and MT cytoskeleton have long been known to be intimately linked, for example during the directional migration of axonal growth cones. Indeed the main molecular linkers connecting actin and microtubules belong to the microtubule plus-end tracking proteins (+TIPs), including EB1 (Coles C. H. and Bradke F. 2015. Coordinating neuronal actin-microtubule dynamics. Curr. Biol.), and a direct association of EB1 with actin has been demonstrated in vitro (Alberico et al. 2016. Interactions between the Microtubule Binding Protein EB1 and F-Actin. J Mol Biol.). Recent studies also proposed a role of actin in the MT spindle formation during mitosis (Farina F. el al 2019. Local actin nucleation tunes centrosomal microtubule nucleation during passage through mitosis. EMBO J). Our results suggest a model where MT dynamics at the wound would permit the association of the +TIPs EB1 with actin that would then facilitate ring closure.

2) Quantitation of data. This manuscript characterizes a new method for wounding and describes the dynamics of various cellular components during the healing process. To make it a more valuable resource for future research, increased quantitation of the data, showing both reproducibility over a number of cells, as well as showing the kinetics of various components relative to each other should be provided.

The formation of an actin ring is a highly reproducible phenotype in our hands. It has become our general control (after the calcium wave) that we use to adjust the 405 laser settings to obtain a normal wound response in control worms. And it has allow us to implement a new wounding platform on a confocal microscope, still using a 405 laser from a FRAP module, in addition to our previous setting on two different spinning disk microscopes. We have now generated several new strains combining the actin reporter with MT binding proteins, and added quantification for actin and EB1 recruitment as mentioned above. In addition to new videos, we now provide kymograph analysis. We also have added the number of worm analysed in the different figures and a detailed source data file reporting for all quantification we have made (Figure 2H, 3B, 3F, 4B, 4C, 4F, 4G, 4H, 5A, 5B, 5C, 6H, 6I, Figure 8—figure supplement 1). Finally, we have added a supplementary figure (Figure 8—figure supplement 1) with quantification of the onset of recruitment and a proposed timeline of events.

There were a number of points that were not entirely clear in the manuscript, such as how microtubules control SNF-12 localization (and further discussed in detail below). Changes to the text to clarify many of these confusions, listed in the reviews below, would strengthen the manuscript.

We have changed the text as explained in more detail below.

Reviewer #1:This manuscript establishes a robust assay for wounding the *C. elegans* lateral epidermis. Using this assay, the authors characterize the membrane and cytoskeletal responses to wounding, which are similar to previously studies Drosophila and Xenopus models. They also show a remarkable increase in the microtubule end-binding protein Eb1, at the wound site. Functionally, they find that disrupting microtubule dynamics inhibits actin accumulation and the normal immune response. There are several really interesting observations in this paper, however, the conclusions at present could be strengthened by increased rigor throughout as different interpretations of the data exist.1) As they are establishing a novel wounding methodology and characterizing the response, increased quantitation of the dynamics and extent of responses in Figures 1-3 are required. For all of these, it would be ideal to have (in addition to the videos shown) quantitations over several videos to see the reproducibility of the responses. This will also be helpful in laying out the temporal order of changes occurring. I find this to be especially important as the authors suggest that the microtubule response occurs before the actin accumulation, though this is not clear from the video presented.

We have added quantifications as mentioned above.

2) Constriction of the F-actin at the wound. This is documented for a single wound in Figure 3H, but is not obvious in the videos provided. It is important to document whether this is a constriction or just a loss of F-actin over time. This is especially confusing as the Introduction states that healing hear is not driven by acto-myosin contraction, but rather by Arp2/3 mediate F-actin assembly.

We provide clearer evidence for the closure of the actin ring in new figures and videos. We indeed believe that the actin ring is not closing by contraction but by inward directed polymerisation of actin. We included the recruitment of Arp2/3 complex protein ARX-2 (Figure 2), confirming the role of branched actin polymerisation in actin closure demonstrated in (Xu and Chisholm, 2011).

3) The accumulation of tubulin at wound sites was dramatically less than Eb1. This is perhaps surprising and should be discussed.

Please see our answer above and to point 1 of Reviewer #2.

Along these lines, it was not clear whether the authors could track Eb1 puncta within the wound site due to density of puncta. Are all puncta motile and analyzable?

It was indeed not possible to track EB1 comets after they reach the wound due to their dense accumulation. But we were able to track the peripheral comets in 9 wounds and 5 control regions. These analyses reveal that their trajectories were not directed toward the wound. We believe that EB1 signal do reflect the dynamics of the + tips MT until they reach the wound (see Figure 2—figure supplement 1). As mentioned above, we have now visualised the recruitment at the wound of another MT + tip protein dynactin 2.

When at the wound, EB1 comets stop, accumulate as a ring that migrates at the same speed as the actin ring to close the wound (see kymographs Figure 2). As we now discuss, these observations are compatible with EB1 binding to actin at the wound.

4) Epistasis of actin and microtubule accumulation. The authors state in the Discussion that Eb1 accumulates before actin. However, this is not clear from the videos provided.

As mentioned above, in addition to new videos, we now provide kymograph analysis. Both show that EB1 associated at the leading edge of the actin while the ring forms and as the ring closes.

Functionally, there is a decrease in F-actin recruitment when tubulin genes are knocked down or microtubules disrupted. However, the timepoints here were late and don't address whether initial accumulation occurs but does not increase or whether it peaks and goes down, or never increases in the first place.

We have added a quantification of the phenotype and show that actin is recruited later and to a lesser extent in *tbb-2* RNAi treated worms (Figure 4).

An additional concern along these lines is that the F-actin as reported by LifeAct appears to be significantly disrupted in the unwounded tubulin mutant epidermis (S2B – to a similar extend as the Act2 mutant). This raises the caveat that it may not be a wound-induced effect, but a general effect on cytoskeletal organization.

We thank the reviewer for highlighting this point that we had reported but not discussed in the original submission. The actin pattern is somewhat affected in young adults after *tbb-2* abrogation during the L4 larval stage. Between L4 and adult, worms molt. There is a major reorganisation of both actin and MT networks during molting (Costa et al., 1997; Lazetic et al., 2018); and the reorganisation of both networks seems to highly coordinated (NP unpublished). Altering MTs could therefore impact the organisation of the actin cytoskeleton. We now mention this briefly in the Discussion. We expect that the coordination of MT and actin networks upon molting or after wounding will use the same mechanisms, and this will be the subject of future studies.

Returning to the reviewer’s concern, within a population of *tbb-2*(RNAi)-treated worms, some worms have an almost normal actin network. If we analyse these worms, we can still observe the concentric wave of actin as in the control but less actin accumulates at the wound site and wound closure is less efficient (Figure 4). High resolution live imaging with a confocal microscope revealed that some MT at the wound site do pattern the growing actin patch (Figure 3C, Video 9). Together with the fact that EB1 is found at the leading edge of the actin ring during its formation and closure, this suggests that MT dynamics facilitate actin accumulation at the wound. We have modified the text accordingly.

5) Connection to the immune response. I felt that I needed more discussion of how the SNF-12 localization connected to the phenotype. Is the localization of the SNF-12 transcription factor at the wound site necessary for its transcriptional activity? How do you imagine this occurs.

Please see our answer to point 2 of Reviewer #2. We regret that we didn’t make clear that SNF-12 is an amino acid transporter that we believe is acting as a signalling protein. We have added to the Discussion “We speculate that this spatial regulation of signalling may contribute to restricting defence gene activation to the immediate vicinity of a wound or site of infection within the major epidermal syncytium.”

Are microtubules required for MAPK kinase activity or DCAR-1 signaling in any way? Authors should rule out the possibility that microtubules affect this signaling axis and that they work through localization of SNF-12.

We apologize if we were not clear enough in the text. DCAR-1 signalling is entirely dependent upon SNF-12, so yes, microtubules affect this signalling axis and they work through localization of SNF-12. We have changed the text in the Discussion to make it clearer. We know that DCAR-1 (the most upstream element of the MAPK pathway described so far) is not recruited to the wound site before SNF-12 but much later, one hour after wounding with a general increased expression. We believe that these observations go beyond the scope of the current paper.

Along these lines, in the induction of AMP's in response to cuticle defects similarly dependent on tubulin genes?

We previously reported that in furrow-less Dpy mutants there is a constitutive high expression of the *nlp* AMP genes (Dodd et al., 2018). We have found that blocking MT dynamics suppresses this high expression (N.P. unpublished). We could add these results if absolutely required but believe they are beyond the scope of the current paper.

6) SNF-12 clusters were significantly lower in unwounded epidermis of tubulin mutants compared to controls. One possibility is that microtubules are required for cluster formation and not localization of SNF-12.

We agree and we now write in the text, “When we disrupted MTs, via RNAi of *tba-2*, SNF-12 patterning and recruitment were severely compromised (Figure 6G-I).” “Together, these results suggest that MTs play an important role in SNF-12 localization and dynamics and thereby in the induction of AMP gene expression.”

Reviewer #2:[…] Several points need to be addressed:1) It is very difficult to tell what, exactly, is happening with the microtubules at the wound site because the different approaches seem to be suggesting different things. That is, in some cell types microtubules accumulate around wounds in a radial array such that many microtubules are focussed toward the wounds (e.g. Mandato and Bement, 2003).. Such an arrangement would be consistent with the accumulation of the EB1-GFP fluorescence around wounds (but see below). However, it is not consistent with the movement of the EB1-GFP comets, which appear to move in all directions and in the absence of any obvious bias toward the wounds. The live imaging of the microtubules themselves (Figure 3C) is also inconsistent with a radial array. As with the movement of the comets, it seems to indicate that the microtubules are randomly oriented with respect to the wound. Now, one might be tempted to imagine that the intense EB1-GFP signal that accumulates around the edge of the wound is sufficient evidence of some kind of wound-localized array of microtubules but there are two problems with this notion. First, that signal is not obviously in comet form, rather, it just appears to be a uniform slathering of GFP on the plasma membrane. Second, EB1 only associates with growing microtubules plus ends; it does not associate with stable microtubule plus ends (e.g. Akhmanova and Steinmetz, 2015). Thus, the apparently stable accumulation of EB1-GFP at the wound edge is rather mysterious. To settle this point the authors need to fix wounded embryos and stain for microtubules or find an alternative way to characterize the organization of microtubules around wounds.

We agree with the reviewer that our observation in the nematode epidermis are not wholly consistent with what has been reported in other systems. To address the reviewer’s point, we chose not to fix the worm as explained above, but have now analysed more MT markers like DNC-2, MAPH-1 and another tubulin isotype TBA-1, in each case generating strains that allow us to visualise at the same time EB1 and/or actin recruitment. Further, we have increased the spatial resolution of our study by implementing the FRAP module for wounding on a confocal microscope, tracked EB1 comets and analysed EB1 and actin ring closure. Together these allow us to have a more complete description of the reorganisation of MTs that accompanies wounding, as explained above.

2) This may reflect my ignorance, but it is not at all clear to me what links the transport of SNF-12 to wounds and the triggering of the innate wound response. Does something special happen to the SNF-12 at wound edges? I was under the impression that SNF-12 exerts its effects on transcriptional control of the response after entering the nucleus as previously reported by the authors (Dierking et al., 2011). While it is not incumbent on the authors to trace out all of the steps in the pathway that leads to an upregulation of the immune transcriptional response, it would be helpful to have a least a little bit of information on why recruitment to the wound is important.

This is a very good question. We do not know how or why the localisation of the amine transporter SNF-12 would affects its signalling. It is known that, to be functional, some signalling complexes need to be assembled on particular vesicular compartments (e.g. TLR7). In our system, SNF-12 localisation could potentially affect its interaction with the Stat transcription factor STA-2.

Understanding this will be a challenge for the future. We have altered the Discussion to address the point explicitly: “The fact that overexpression in the epidermis of the MT severing protein Spastin caused a similar effect on SNF-12 dynamics as knocking down tubulin expression, supports a role for MT in positioning SNF-12-containing vesicles. […] We speculate that this spatial regulation of signalling may contribute to restricting defence gene activation to the immediate vicinity of a wound or site of infection within the major epidermal syncytium.”